# T-cell-derived IFN-γ suppresses T follicular helper cell differentiation and antibody responses

Eleonora Sala[1,2,9], Maria Nelli [1,2,9], Chiara Laura[1,2,3], Pietro Di Lucia[1,2], Cristian Gabriel Beccaria [2], Elisa B Bono[2], Marta Mangione[1,2], Davide Marotta[1,2], Valentina Sperto [1,2], Marta Grillo [1,2], Leonardo Giustini[2], Fabio Tosi [1,2], Jia Nie[4], Daehong Kim [5], Giuliana Furiato[1,2], Chiara Malpighi[2], Eleonora Consolo[1,2], Burkhard Becher[5], Eyal David[6], Merav Cohen[7], Amir Giladi[6], Ido Amit[6], Remy Bosselut[4], Luca G Guidotti [1,2], Matteo Iannacone [1,2,8,10]✉ & Mirela Kuka [1,2,10]✉

## Abstract

CD4[+] T cells play a critical role in antiviral humoral and cellular immune responses. We have previously reported that subcutaneous lymphocytic choriomeningitis virus (s.c. LCMV) infection is characterized by a stark compartmentalization of CD4[+] T cells, leading to strong $T_H1$ cell polarization but virtually absent T follicular helper ($T_{FH}$) cells, key drivers of humoral immunity. Here, we investigate the mechanisms responsible for this impaired $T_{FH}$ differentiation. We show that T-bet[+] cells induced by LCMV infection encompass a $T_H1$ cell subset expressing granzyme B (GzmB), and a Tcf-1[+] cell subset that retains the potential for $T_{FH}$ differentiation without expressing mature $T_{FH}$ markers. Notably, IFN-γ blockade enables full differentiation of Tcf-1[+] cells into $T_{FH}$ cells, formation of germinal centers, and increased antibody production. Suppression of $T_{FH}$ cells by IFN-γ is not directly mediated by CD4[+] T cells but rather involves another cell type, likely dendritic cells (DCs). Our study provides novel insights into the mechanisms underlying early CD4[+] T-cell polarization and humoral responses to viruses, with the potential to facilitate the development of effective vaccine strategies.

**Keywords** CD4[+] T Cells; Viral Infection; IFN-γ; B-cell Responses
**Subject Categories** Immunology; Microbiology, Virology & Host Pathogen Interaction

## Introduction

CD4[+] T cells play a crucial role in orchestrating adaptive immune responses against pathogens, guiding a complex array of signals and differentiation processes. Following their priming in secondary lymphoid organs, antigen-specific CD4[+] T cells undergo both clonal expansion and differentiation into effector cells (Zhu et al, 2010). Throughout this process, these T cells are exposed to a diverse range of cytokines from infected cells, dendritic cells (DCs), and stromal cells. In response, CD4[+] T cells embark on specific differentiation pathways, leading to the formation of distinct T helper cell subsets (Mempel et al, 2004; Walsh and Mills, 2013; Eisenbarth, 2018; Tuzlak et al, 2021).

Infection by viruses or intracellular bacteria primarily leads to the generation of $T_H1$ and $T_{FH}$ cells (Sheikh and Groom, 2020; Kuka and Iannacone, 2021). $T_H1$ cells, characterized by the expression of the master transcription factor T-bet and the production of high levels of IFN-γ, promote macrophage activation and bolster CD8[+] T cell responses (Szabo et al, 2000; Schoenborn and Wilson, 2007; Sercan et al, 2010; Snell et al, 2016). Furthermore, autocrine IFN-γ plays a role in promoting the expansion and maintaining the $T_H1$ phenotype (Bradley et al, 1996; Lighvani et al, 2001; Whitmire et al, 2005). Conversely, $T_{FH}$ cells, which express Bcl-6 and CXCR5, migrate to B cell follicles, where they interact specifically with cognate B cells, facilitating germinal center reactions and the subsequent generation of high-affinity, class-switched antibodies (Crotty, 2011; Vinuesa et al, 2016).

Ideally, $T_H1$ and $T_{FH}$ subsets coexist, each contributing to adaptive immune responses by predominantly supporting cellular or humoral immunity, respectively. Based on literature, the bifurcation between $T_{FH}$ and $T_H1$ fates seems to take place within a few days upon CD4[+] T cell activation (Choi et al, 2011; DiToro et al, 2018). However, previous work has shown some degree of overlap or competition between these two CD4[+] T cell subsets (Nakayamada et al, 2011; Lönnberg et al, 2017). For example, it was reported that $T_{FH}$ and $T_H1$ share a transitional phase expressing both T-bet and Bcl-6. While the cells progress into reinforcing $T_H1$ phenotype, T-bet suppresses further $T_{FH}$ differentiation by competing with Bcl-6. This competition seems to be cell-intrinsic since CD4[+] T cells lacking T-bet differentiate into $T_{FH}$ (Nakayamada et al, 2011; Lönnberg et al, 2017). In another study,

[1]School of Medicine, Vita-Salute San Raffaele University, Milan, Italy. [2]Division of Immunology, Transplantation, and Infectious Diseases, IRCCS San Raffaele Scientific Institute, Milan, Italy. [3]Center for Omics Sciences, IRCCS San Raffaele Scientific Institute, Milan, Italy. [4]Laboratory of Immune Cell Biology, Center for Cancer Research, National Cancer Institute, National Institutes of Health, Bethesda, MD, USA. [5]Institute of Experimental Immunology, University of Zurich, Zurich, Switzerland. [6]Department of Immunology, Weizmann Institute of Science, Rehovot, Israel. [7]Department of Clinical Microbiology and Immunology, Faculty of Medical and Health Sciences, Tel Aviv University, Tel Aviv, Israel. [8]Experimental Imaging Centre, IRCCS San Raffaele Scientific Institute, Milan, Italy. [9]These authors contributed equally: Eleonora Sala, Maria Nelli. [10]These authors contributed equally: Matteo Iannacone, Mirela Kuka. ✉E-mail: iannacone.matteo@hsr.it; kuka.mirela@hsr.it

Bcl6-expressing $T_{FH}$ cells generated upon viral infection expressed T-bet, which was critical for their development and function and transcriptionally required for proper $T_{FH}$ cell programming (Weinstein et al, 2018). However, T-bet expression by $T_{FH}$ has also been reported to render these cells dysfunctional, like in severe malaria infection (Obeng-Adjei et al, 2015; Ryg-Cornejo et al, 2016; Hansen et al, 2017). In this context, concomitant blockade of IFN-γ and TNF-α resulted in enhanced $T_{FH}$ differentiation and improved antibody responses (Ryg-Cornejo et al, 2016). Thus, the interplay between $T_{H}1$ and $T_{FH}$ is marked by controversy and needs to be further dissected.

We have recently reported that subcutaneous (s.c.) infection with lymphocytic choriomeningitis virus (LCMV) results in almost exclusive $T_{H}1$ differentiation and impaired $T_{FH}$ induction (De Giovanni et al, 2020). While this observation is in line with the strong cellular responses and the weak neutralizing antibody (nAb) responses triggered by non-cytopathic viruses such as LCMV (Hangartner et al, 2006), the precise cellular and molecular mechanisms influencing this bias remain elusive.

In this study, we uncovered the heterogeneity within LCMV-induced T-bet⁺ cells, identifying two distinct subsets: a Tcf-1⁺ subset and a granzyme B (GzmB)⁺ subset. Surprisingly, neither subset required the canonical $T_{H}1$-polarizing cytokine IL-12 for differentiation. Instead, IFN-γ emerged as a key regulator, driving the expansion of GzmB⁺ cells while suppressing the maturation of Tcf-1⁺ cells into fully differentiated $T_{FH}$. Notably, blocking IFN-γ restored the $T_{FH}$ population and enhanced germinal center B cells and Ab production. These findings reveal novel mechanisms that limit $T_{FH}$ differentiation and Ab production during viral infections, providing valuable insights for developing innovative vaccination strategies.

# Results

## Heterogeneity of T-bet⁺ CD4⁺ T cells upon LCMV infection

In aiming to understand the determinants responsible for the impaired $T_{FH}$ differentiation upon s.c. LCMV infection, our initial step was the comprehensive transcriptional profiling of antigen-specific CD4⁺ T cells. We utilized single-cell RNA sequencing (scRNA-seq) to analyze the diverse transcriptomic landscapes. To this end, naive Smarta CD4⁺ T cells, which are reactive to the MHC-II-restricted GP61-80 epitope of the LCMV glycoprotein (Oxenius et al, 1998), were adoptively transferred into wild-type (WT) mice. This was done one day before the mice were subjected to s.c. (intrafootpad) infection with rLCMV (a recombinant LCMV clone 13 expressing the LCMV WE glycoprotein recognized by Smarta TCR-transgenic cells (Fallet et al, 2016; De Giovanni et al, 2020). Five days post-infection, a time point characterized by notable expansion of Ag-specific CD4⁺ T cells (De Giovanni et al, 2020), we performed the isolation and FACS-based sorting of Smarta CD4⁺ T cells from footpad-draining popliteal lymph nodes (dLNs) (Fig. 1A). Consistent with our previous report (De Giovanni et al, 2020), LCMV-specific CD4⁺ T cells generated in this setting predominantly expressed T-bet, a key transcriptional regulator of $T_{H}1$ differentiation, while $T_{FH}$ marker CXCR5 was notably absent (Appendix Fig. S1). For comparative analysis, Smarta CD4⁺ T cells from mice infected with rVSV, a recombinant VSV that expresses the LCMV WE glycoprotein and induces $T_{FH}$

differentiation (Fallet et al, 2016; De Giovanni et al, 2020), served as a control (Appendix Fig. S1).

We then performed massively parallel single-cell RNA-sequencing (MARS-seq) on QC-positive single T cells. Using the Seurat R package (Stuart et al, 2019), we scrutinized the scRNA-seq dataset, revealing distinct cellular clusters through uniform manifold approximation and projection (UMAP) analysis (McInnes et al, 2018). It was evident that control Smarta CD4⁺ T cells isolated from naive mice ($n = 369$, marked in blue) were characterized by a higher expression of canonical naive T cell markers such as *Ccr7*, *Sell*, and *Klf2* (Appendix Fig. S2A). In stark contrast, the CD4⁺ T cells from the infected hosts were discretely clustered based on the infective agent, confirming the considerable divergence between the CD4⁺ T cell subsets post VSV or LCMV infection (Fig. 1B; Appendix Fig. S2B and Dataset EV1) (De Giovanni et al, 2020).

Focusing on the rLCMV condition, we discerned two distinct clusters within the Smarta CD4⁺ T cells (Fig. 1C). The dominant cluster 0 ($n = 557$) showed upregulated expression of genes such as *Gzmb*, *Nkg7*, and *Ly6c2* (Fig. 1D,E and Dataset EV2). Conversely, cluster 1 ($n = 86$) was characterized by elevated levels of *Tcf7*, a transcription factor implicated in $T_{FH}$ differentiation (Xu et al, 2015) (Fig. 1D,E and Dataset EV2). Intriguingly, cluster 1 overlaid with cells at the rLCMV and rVSV interface when backgated onto the original UMAP, hinting at a transitional phenotype (Appendix Fig. S2C). In addition, a GSEA analysis showed that the rVSV signature was significantly enriched in cluster 1 but not in cluster 0 (Appendix Fig. S2D), indicating that cluster 1 is enriched in cells with a transcriptional signature similar to the $T_{FH}$ in VSV infection.

Flow cytometry analyses were performed to corroborate the scRNA-seq findings, revealing two distinct populations within the T-bet⁺ CD4⁺ Smarta T cells five days after rLCMV infection. These were classified based on the expression of GzmB and Tcf-1 (encoded by *Tcf7*) proteins, paralleling the scRNA-seq identified subsets (Fig. 1F,G; Appendix Fig. S3A–C). The GzmB⁺ population was predominant, while the Tcf-1⁺ subset constituted a smaller, yet significant fraction. Further examination of the GzmB⁺ cells showed elevated CXCR6 and Ly6C expression, along with marginally increased T-bet levels compared to the Tcf-1⁺ group (Appendix Fig. S3D). The Tcf-1⁺ cells instead showed slightly higher levels of Bcl-6, CXCR5, and CXCR3 (Appendix Fig. S3D). Also the endogenous CD4⁺ T cell compartment seven days post rLCMV infection contained these two subsets, with comparable frequencies to the Smarta cells (Fig. 1H,I; Appendix Fig. S4A,B). Functionally, the GzmB⁺ subset distinguished itself with a robust IFN-γ response upon GP61 peptide re-stimulation, a trait not as pronounced in the Tcf-1⁺ cells (Fig. 1J). The differential IFN-γ expression was further quantified, revealing significantly higher mean fluorescence intensity (MFI) in GzmB⁺ cells compared to their Tcf-1⁺ counterparts (Fig. 1K).

These findings underscore the existence of two functionally and phenotypically distinct subsets within the T-bet⁺ CD4⁺ T cell population upon s.c. rLCMV infection: the GzmB⁺ population probably representing the classical IFN-γ-producing $T_{H}1$ cells, and the Tcf-1⁺ population as a putative precursor of bona fide $T_{FH}$.

## IFN-γ suppresses T follicular helper differentiation and antibody production

Next, we sought to elucidate the molecular determinants driving the $T_{H}1$ polarization and heterogeneity induced by LCMV

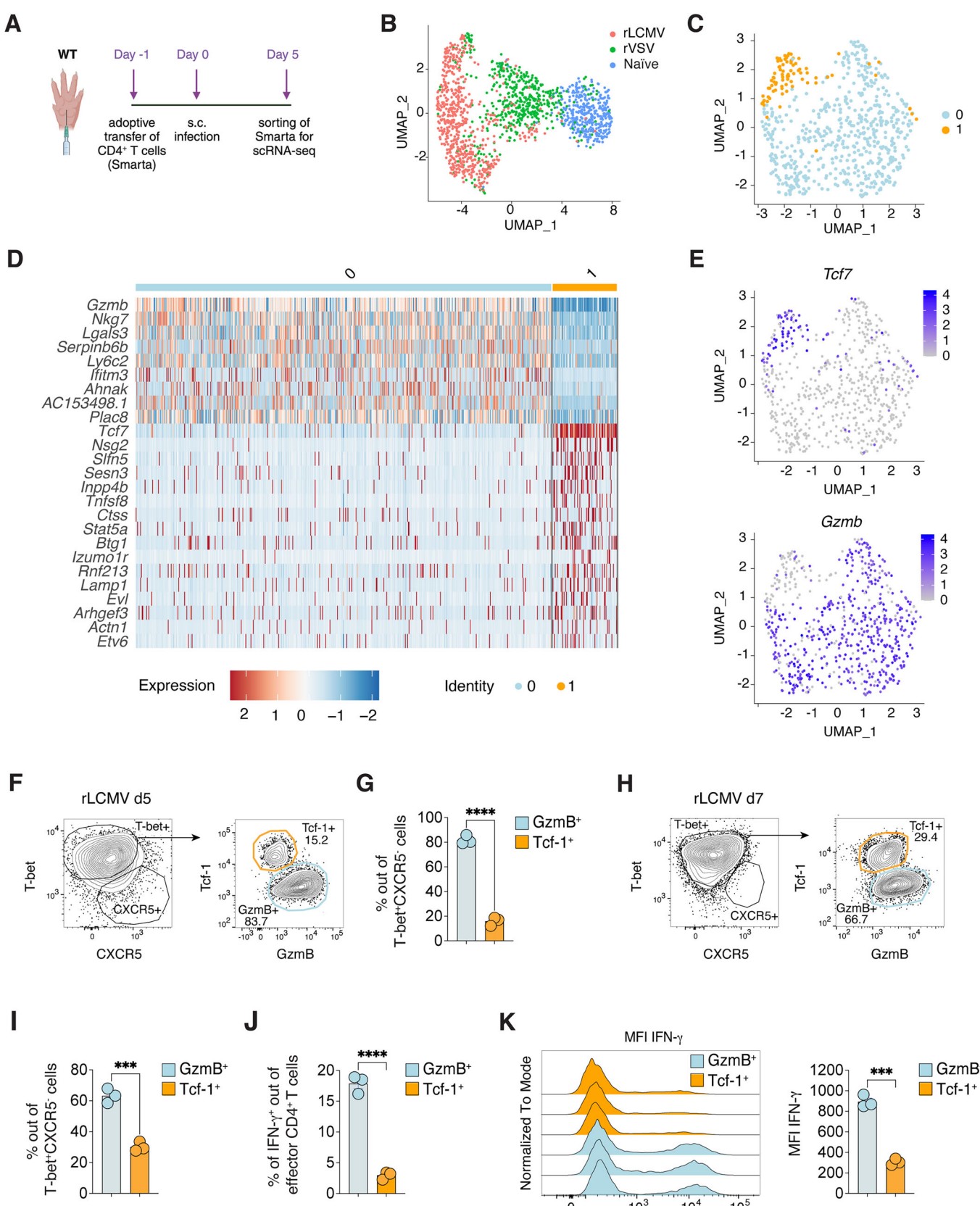

**Figure 1. Heterogeneity of T-bet$^+$ CD4$^+$ T cells upon LCMV infection.**

(A) Schematic representation of the experimental setup for the results described in B-E. 0.5*10$^6$ purified Ag-specific CD45.1$^+$ CD4$^+$ T cells (Smarta) were transferred into CD45.2$^+$ WT recipients one day before s.c. infection with either rLCMV or rVSV (1*10$^5$ FFU or PFU/footpad). Smarta CD4$^+$ T cells were FACS-sorted from the dLNs five days upon infection based on CD45.1 expression. (B) UMAP projection of sorted cells. Each dot corresponds to a single cell, colored according to different samples. rLCMV cells (red, 643 cells), rVSV cells (green, 498 cells), naive Smarta CD4$^+$ T cells (light blue, 369 cells). (C) UMAP projection of Smarta CD4$^+$ T cells sorted from rLCMV-infected mice. Each dot corresponds to a single cell, colored according to the unbiased clusters identified: cluster 0 (light blue, 557 cells) and cluster 1 (orange, 86 cells). (D) Heatmap of normalized and scaled expression values of the top marker genes identifying the two clusters (logFC threshold: ±1 and adjusted $p$-value < 0.05 filters were applied). Color coding of the bar on the top of the heatmap as in (C). (E) Feature plot representation of the expression level of the natural-log normalized expression level of *Gzmb* and *Tcf7* on the scRNA-seq subset described in (C). (F) 0.5*10$^6$ purified CD45.1$^+$ Smarta CD4$^+$ T cells were transferred into CD45.2$^+$ WT recipients 1 day before s.c. rLCMV infection (1*10$^5$ FFU/footpad). dLNs were analyzed 5 days post infection. Representative flow cytometry plots showing the frequencies of Tcf-1$^+$ and GzmB$^+$ cells among T-bet$^+$CXCR5$^-$ Smarta CD4$^+$ T cells in the dLNs. Numbers represent the percentage of cells within the indicated gate. (G) Quantification of GzmB$^+$ and Tcf-1$^+$ cells expressed as percentages of T-bet$^+$CXCR5$^-$ Smarta CD4$^+$ T cells in dLNs of mice described in (F). $n = 3$. Mean ± SEM is shown. Data are representative of at least five independent experiments. An unpaired two-tailed t test was applied. ****$p$-value = 0.00002090487. (H) CD45.2$^+$ WT mice were infected s.c. with rLCMV (1*10$^5$ FFU/footpad) and dLNs were analyzed 7 days post infection. Representative flow cytometry plots showing Tcf-1$^+$ and GzmB$^+$ cells among T-bet$^+$CXCR5$^-$ endogenous CD4$^+$ T cells in the dLNs. Numbers represent the percentage of cells within the indicated gate. (I) Quantification of GzmB$^+$ and Tcf-1$^+$ cells expressed as percentages of T-bet$^+$CXCR5$^-$ endogenous CD4$^+$ T in dLNs. $n = 3$. Mean ± SEM is shown. Data are representative of three independent experiments. An unpaired two-tailed t test was applied: ***$p$-value = 0.0005194870. (J) Quantification of IFN-γ$^+$ cells among the GzmB$^+$ and Tcf-1$^+$ subsets out of effector CD44$^+$ CD62L$^{lo}$ CD4$^+$ T cells in dLNs of mice described in (H). $n = 3$. Mean ± SEM is shown. Data are representative of three independent experiments. An unpaired two-tailed t test was applied: ****$p$-value = 0.0000829963. (K) Representative flow cytometry plot (left panel) and quantification (right panel) showing the fluorescence intensity of IFN-γ expression within GzmB$^+$ or Tcf-1$^+$ CD4$^+$ T cells of mice described in (H). $n = 3$. Mean ± SEM is shown. Data are representative of three independent experiments. An unpaired two-tailed t test was applied: ***$p$-value = 0.0001170575. Source data are available online for this figure.

infection. We initially investigated the role of IL-12, a well-known T$_H$1-polarizing cytokine (Heufler et al, 1996; Athie-Morales et al, 2004). In line with previous studies on viral infections (Schijns et al, 1998; Oxenius et al, 1999; Krueger et al, 2021), neutralization of IL-12 did not significantly alter CD4$^+$ T cell differentiation (Appendix Fig. S5A–D), prompting us to explore the influence of another cytokine known for reinforcing T$_H$1 identity through positive feedback loops, IFN-γ (Wakil et al, 1998; Lighvani et al, 2001; Schulz et al, 2009).

We examined the effects of IFN-γ by transferring Smarta CD4$^+$ T cells into wild-type mice treated with IFN-γ blocking antibodies (Fig. 2A). IFN-γ blockade affected the expansion of Smarta CD4$^+$ T (Appendix Fig. S6A), in line with the previously reported role for IFN-γ in expansion of antiviral CD4$^+$ T cells (Whitmire et al, 2005). Intriguingly, blocking IFN-γ shifted the balance in favor of the Tcf-1$^+$ subset at the expense of the GzmB$^+$ population, both in terms of percentages and absolute numbers (Fig. 2B; Appendix Fig. S6B). This was accompanied by an upregulation of the canonical T$_{FH}$ marker CXCR5 in the Tcf-1$^+$ subset (Fig. 2C,D). In addition, Tcf-1$^+$ cells showed lower levels of T-bet and higher levels of PD-1, Bcl-6, CD95 and CXCR3 when IFN-γ was blocked (Appendix Fig. S6C). As a result, IFN-γ blockade resulted in a decrease in T-bet$^+$ T$_H$1 cells while promoting the emergence of CXCR5$^+$ T$_{FH}$ cells (Fig. 2E,F; Appendix Fig. S6D). Notably, the effect of IFN-γ blockade in CD4$^+$ T cell polarization was not a result of higher viral replication, since viral titers in mice receiving the α-IFN-γ neutralizing Ab were lower at day 3 and similar to control mice at day 5 after LCMV infection (Appendix Fig. S6E). In addition, we have previously shown that CD4$^+$ T cells differentiate into T$_H$1 but not T$_{FH}$ even when animals are infected with 100-fold higher or lower viral loads (De Giovanni et al, 2020).

To delve deeper into the transcriptional changes induced by IFN-γ, we conducted single-cell RNA sequencing (scRNA-seq) on Smarta cells from mice treated with either PBS or anti-IFN-γ antibody. This analysis unveiled six distinct clusters within LCMV-specific CD4$^+$ T cells (Fig. 2G; Appendix Fig. S7A and Dataset EV3). Among these, five expressed *Gzmb*, while one cluster (cluster 5) was *Gzmb*-negative but displayed high levels of *Tcf7* (Fig. 2H and

Dataset EV3). *Gzmb$^+$* clusters showed lower *Cxcr5* expression and higher *Tbx21* (which encodes for T-bet) levels (Appendix Fig. S7B), consistent with previous protein expression findings (Appendix Fig. S3D). These *Gzmb$^+$* cells also exhibited slightly higher expression of *Ifngr1*, suggesting heightened sensitivity to IFN-γ (Appendix Fig. S7C). Notably, cluster 0, characterized by *Gzma* expression, was almost exclusively present in PBS-treated animals, indicating a strong dependence on IFN-γ for its development (Fig. 2I,J; Appendix Fig. S7D).

Further analysis of the *Tcf7*-expressing cluster revealed a significant enrichment of cells from the anti-IFN-γ-treated group, constituting 71% of the cluster, compared to 29% from PBS-treated mice. This confirms our observation of Tcf-1$^+$ cell increase following IFN-γ blockade (Fig. 2I; Appendix Fig. S6B). Additionally, Tcf7-expressing cells in the anti-IFN-γ group displayed markedly higher levels of the gene signature characteristic of cluster 5 compared to all other clusters (Fig. 2K). This finding suggests that the T$_{FH}$ precursor gene signature in LCMV-infected mice is amplified in the absence of IFN-γ.

All in all, our data reveal substantial heterogeneity among T-bet$^+$ cells in the context of LCMV infection. Specifically, one cluster's development relies exclusively on IFN-γ, while the *Tcf7*-expressing cluster is numerically and phenotypically suppressed by IFN-γ, highlighting the complexity of CD4$^+$ T cell differentiation in this scenario.

We further investigated whether Tcf-1$^+$ CXCR5$^+$ cells, generated in the absence of IFN-γ, functionally resemble T$_{FH}$ cells. Confocal microscopy of dLNs from rLCMV-infected mice confirmed a marked increase in Tcf-1$^+$ antigen-specific CD4$^+$ T cells (Smarta) without IFN-γ (Fig. 3A,B). Notably, these Tcf-1$^+$ cells were located not only in interfollicular regions but also within B cell follicles, indicating their transition to fully differentiated and functional CXCR5$^+$ T$_{FH}$ cells (Fig. 3A–C). This observation strongly suggests that IFN-γ inhibits T$_{FH}$ cell differentiation by preventing the maturation of Tcf-1$^+$ cells into CXCR5$^+$ T$_{FH}$ cells.

Interestingly, IFN-γ blockade resulted in a notable rescue of T$_{FH}$ cells even in an endogenous setting where T cells were analyzed seven days upon s.c. infection (Fig. 4A,B). These T$_{FH}$ generated

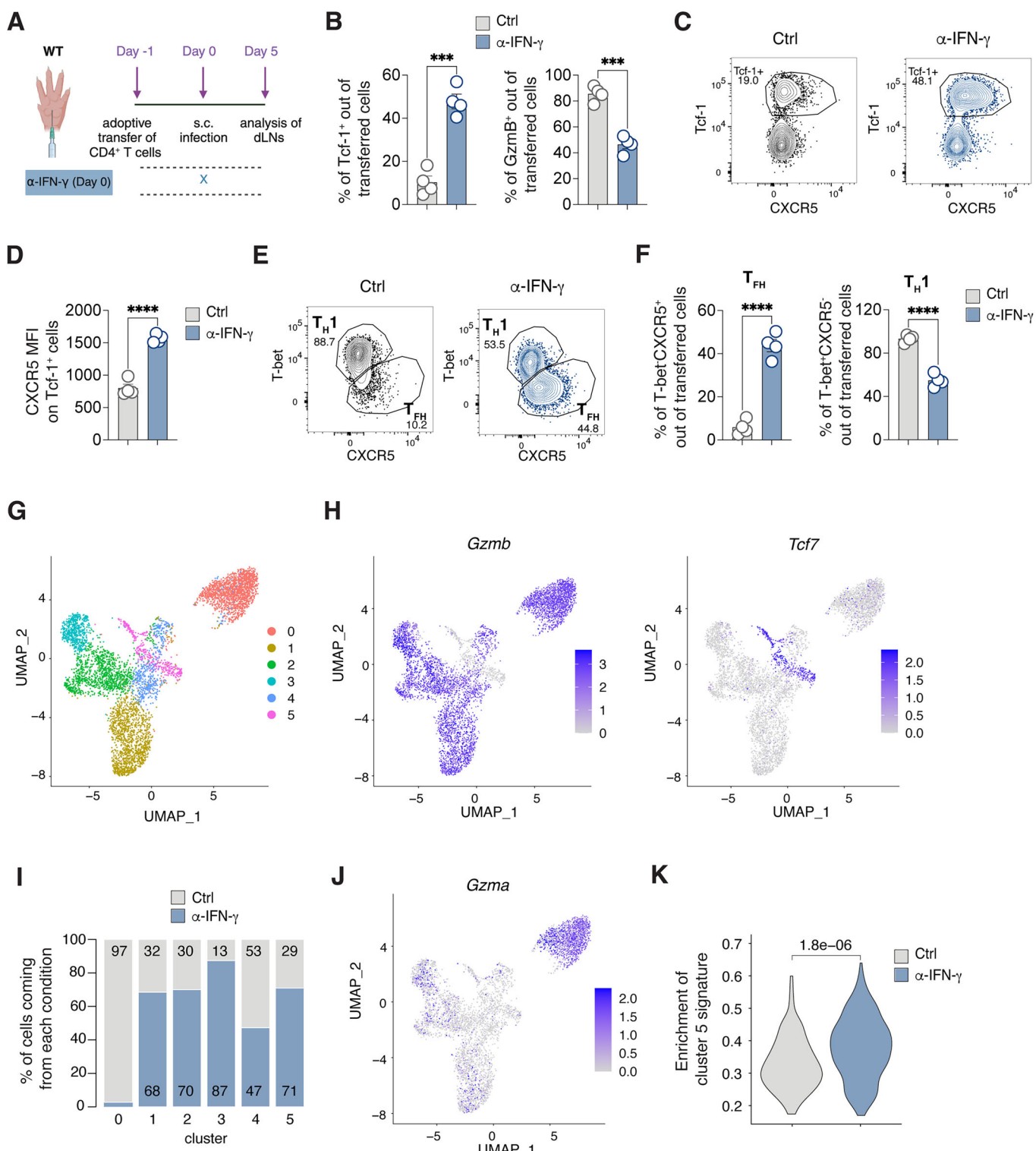

upon IFN-γ blockade expressed Bcl-6 hinting to their complete differentiation (Fig. 4C,D). Moreover, IFN-γ was found to significantly impact B cell activation (Fig. 4E,F). Approximately 15–20% of B cells in the dLNs of rLCMV-infected mice expressed high levels of T-bet seven days post-infection, compared to a meager 1% expressing Bcl-6. Blocking IFN-γ, however, led to an increase in Bcl-6+ B cells and a decrease in T-bet+ B cells (Fig. 4E,F). This shift in B cell populations was in line with higher levels of total LCMV GP-binding IgG antibodies in the absence of IFN-γ (Fig. 4G). Collectively, these findings underscore the inhibitory role of IFN-γ in the development of T_FH cells, germinal center B cells, and antibody responses during viral infection.

**Figure 2. IFN-γ suppresses T follicular helper cell differentiation.**

(A) 0.5*10⁶ purified CD45.1⁺ Smarta CD4⁺ T cells were transferred into CD45.2⁺ WT recipients 1 day before s.c. rLCMV infection (1*10⁵ FFU /footpad). CD45.2⁺ WT recipient mice were also treated with α-IFN-γ blocking antibody (or isotype Ctrl) at day 0. dLNs were analyzed 5 days post infection. (B) Quantification of Tcf-1⁺ (left) and GzmB⁺ (right) in dLNs of mice described in (A) expressed as percentages out of transferred Smarta CD4⁺ T cells. $n = 4$. Mean ± SEM is shown. Data are representative of at least three independent experiments. An unpaired two-tailed t test was applied. ***p-value = 0.0001610861 (Tcf-1⁺) and 0.0001204432 (GzmB⁺). (C) Representative flow cytometry plots showing CXCR5 expression on Tcf-1⁺ and Tcf-1⁻ Smarta CD4⁺ T cells in dLNs of mice described in (A). Numbers represent the percentage of cells within the indicated gate. (D) Quantification of the MFI of CXCR5 on Tcf-1⁺ Smarta CD4⁺ T cells in dLNs of mice described in (A). $n = 4$. Mean ± SEM is shown. Data are representative of at least three independent experiments. An unpaired two-tailed t test was applied. ****p-value = 0.0000338903. (E) Representative flow cytometry plots showing $T_H1$ (T-bet⁺CXCR5⁻) and $T_{FH}$ (T-bet⁻CXCR5⁺) cells among Smarta CD4⁺ T cells in dLNs of mice described in (A). Numbers represent the percentage of cells within the indicated gate. (F) Quantification of $T_{FH}$ (left) and $T_H1$ (right), expressed as percentages out of transferred Smarta CD4⁺ T cells in dLNs of mice described in (A). $n = 4$. Mean ± SEM is shown. Data are representative of at least three independent experiments. An unpaired two-tailed t test was applied. ****p-value = 0.0000272354 ($T_{FH}$) and 0.0000291579 ($T_H1$). (G) UMAP projection of 5746 sorted and sequenced LCMV-specific CD4⁺ T cells. Each dot corresponds to a single cell, colored according to unbiased clusters identified using the Louvain algorithm. (H) Feature plot representation of the natural-log normalized expression level of *Gzmb* (left panel) and *Tcf7* (right panel) on the dataset in (G). (I) Barplot representation of the frequencies of the two conditions (control in gray and α-IFN-γ in blue) in each cluster. (J) Feature plot representation of the natural-log normalized expression level of *Gzma* on the dataset in (G). (K) Violin plot showing the enrichment of a signature composed by marker genes of cluster 5 versus all the other clusters, comparing control (gray) and α-IFN-γ (blue) cells in cluster 5 only. Two-tailed Mann-Whitney test has been performed and the resulting *p*-value is shown. Source data are available online for this figure.

## Timing of IFN-γ-mediated suppression of $T_{FH}$ differentiation

To elucidate the molecular mechanisms and timing of IFN-γ-mediated suppression of $T_{FH}$ differentiation, we explored the effects of IFN-γ blockade initiated at different stages post-infection (Fig. 5A). Blocking IFN-γ at day 3 post-infection did not significantly alter the frequencies of Tcf-1⁺ and GzmB⁺ populations (Fig. 5B). Moreover, while IFN-γ inhibition at day 0 significantly increased CXCR5 expression on Tcf-1⁺ cells, initiating blockade at day 3 produced only minimal, non-significant changes (Fig. 5C; Appendix Fig. S8A). Consequently, an increase in functional $T_{FH}$ frequencies was observed only when IFN-γ blockade was implemented from the onset of infection (Fig. 5D). This suggests that IFN-γ produced in the initial days post-infection is crucial for impeding $T_{FH}$ differentiation.

Further analyses were conducted to ascertain if CD4⁺ T cell polarization influenced by IFN-γ could be observed earlier than day 5 post-infection. Upon examining the LCMV-specific CD4⁺ T cells at day 3 post s.c. infection, we noted an increase in the Tcf-1⁺ population coupled with a decrease in GzmB⁺ cells when IFN-γ was blocked (Fig. 5E,F; Appendix Fig. S8B). These observations indicate that IFN-γ begins shaping CD4⁺ T cell differentiation early during infection, influencing both $T_{FH}$ and $T_H1$ precursor populations. Interestingly, at this early stage, LCMV-specific CD4⁺ T cells were already capable of producing IFN-γ upon antigen re-stimulation, indicating their functional maturity just three days post-infection (Fig. 5G). The production of IFN-γ was reduced in mice treated with the IFN-γ-blocking antibody, aligning with its role in stabilizing the $T_H1$ phenotype (Fig. 5G; Appendix Fig. S8C). Although both T cell subsets could produce IFN-γ, most of the cytokine was derived from GzmB⁺ cells (Fig. 5H).

We next asked for how long could differences in T cell polarization upon IFN-γ blockade be detected. To this end, Smarta CD4⁺ T cells were analyzed 14 days upon infection (Fig. 5I). The frequency of Smarta cells at this time-point was highly variable and significantly lower than day five, due to the contraction phase all T cells undergo after viral clearance (Fig. 5J). Nonetheless, IFN-γ blockade still resulted in increased $T_{FH}$ frequencies (Fig. 5K) indicating that a single-dose of blocking antibody at the beginning of infection was sufficient to affect T cell polarization for weeks.

Overall, our findings demonstrate that the presence of IFN-γ during early priming events crucially influences CD4⁺ T cell differentiation, favoring $T_H1$ cell development at the expense of $T_{FH}$ cells.

## IFN-γ produced by T cells is key in $T_{FH}$ suppression

To investigate which might be the cellular source of the early IFN-γ crucial for $T_{FH}$ suppression, we utilized the IFN-γ-Yellow Fluorescent Protein (YFP) mouse reporter model (Reinhardt et al, 2009) to identify cells expressing this cytokine during the initial stages of LCMV infection. Analysis of IFN-γ-YFP mice revealed diverse immune cell types (identified through gating strategy in Appendix Fig. S9) expressing IFN-γ as early as 24 h post-infection, including NK1.1⁺ group 1 innate lymphoid cells (ILCs) (comprising both NK cells and ILC1s), T lymphocytes, monocytes, and DCs (Fig. 6A, left panel). By 48 h, group 1 ILCs emerged as the predominant IFN-γ-expressing subset, followed by CD8⁺ T cells (Fig. 6A, middle panel). Three days post-infection, group 1 ILCs, CD8⁺ T cells, CD4⁺ T cells, and monocytes were the main IFN-γ producers (Fig. 6A, right panel).

To explore the significance of IFN-γ produced by group 1 ILCs in CD4⁺ T cell polarization, we adoptively transferred Smarta CD4⁺ T cells into WT mice that were treated with neutralizing antibodies against IFN-γ or NK1.1, or a combination of both. Five days after LCMV infection group 1 ILCs were still efficiently depleted (Appendix Fig. S10A). In line with previous experiments, IFN-γ blocking led to a shift towards the $T_{FH}$ cell subset (Fig. 6B,C) with a substantial increase of the Tcf-1⁺ population, a relative decrease of GzmB⁺ cells, and an increase of CXCR5 levels on Tcf-1⁺ cells (Appendix Fig. S10B,C). Notably, treatment with anti-NK1.1 antibody alone did not impact the polarization of T helper cell subsets. Moreover, the polarization pattern in mice treated with the combination of both antibodies mirrored the one observed with anti-IFN-γ alone (Fig. 6B,C; Appendix Fig. S10B,C). Taken together, these findings suggest that, at least in this infection setting, IFN-γ derived from group 1 ILCs is not directly involved in CD4⁺ T cell polarization.

With group 1 ILCs ruled out, we explored other potential sources of IFN-γ. Monocytes and DCs, detected to produce IFN-γ 24 h post-infection (Fig. 6A), were examined. Focusing on DCs, we employed a mixed bone marrow chimera model combining CD11c-DTR (for DC depletion) and IFN-γ⁻/⁻ mice. Depletion of WT DCs,

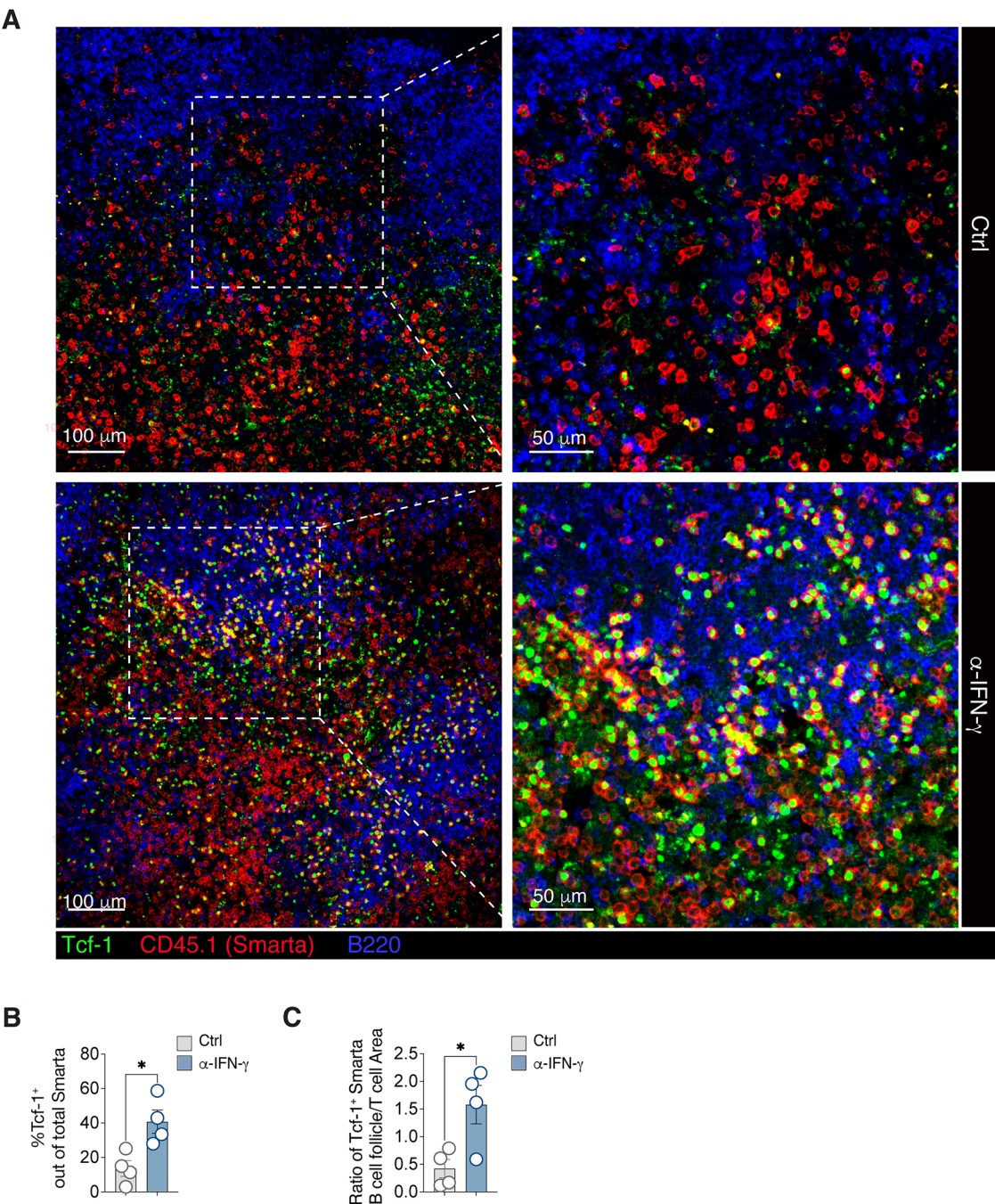

**Figure 3. IFN-γ suppresses T follicular helper cell responses upon viral infection.**

(A) 0.5*10^6 purified CD45.1+ Smarta CD4+ T cells were transferred into CD45.2+ WT recipients one day before s.c. rLCMV infection (1*10^5 FFU /footpad). CD45.2+ WT recipient mice were also treated with α-IFN-γ blocking antibody (or isotype Ctrl) at day 0. dLNs were analyzed by confocal microscopy 5 days post infection. CD45.1+ (Smarta CD4+ T cells) are depicted in red, Tcf-1+ cells in green and B220+ cells (B cell follicles) in blue. Images on the right panels are a magnification of the dotted square areas on the left. Data are representative of two independent experiments. Scale bars represent 100 (left) and 50 (right) μm. (B) Quantification of Tcf-1+ cells out of total Smarta CD4+ T cells in the images (left panels) in (A). n = 4. Mean ± SEM is shown. Data are representative of two independent experiments. An unpaired two-tailed t test was applied. *p-value = 0.0157. (C) Calculation of the ratio of the density of Tcf-1+Smarta+ in the B cell follicle versus the T cell area in the images (left panels) in (A). n = 4. Mean ± SEM is shown. Data are representative of two independent experiments. An unpaired two-tailed t test was applied. *p-value = 0.0237. Source data are available online for this figure.

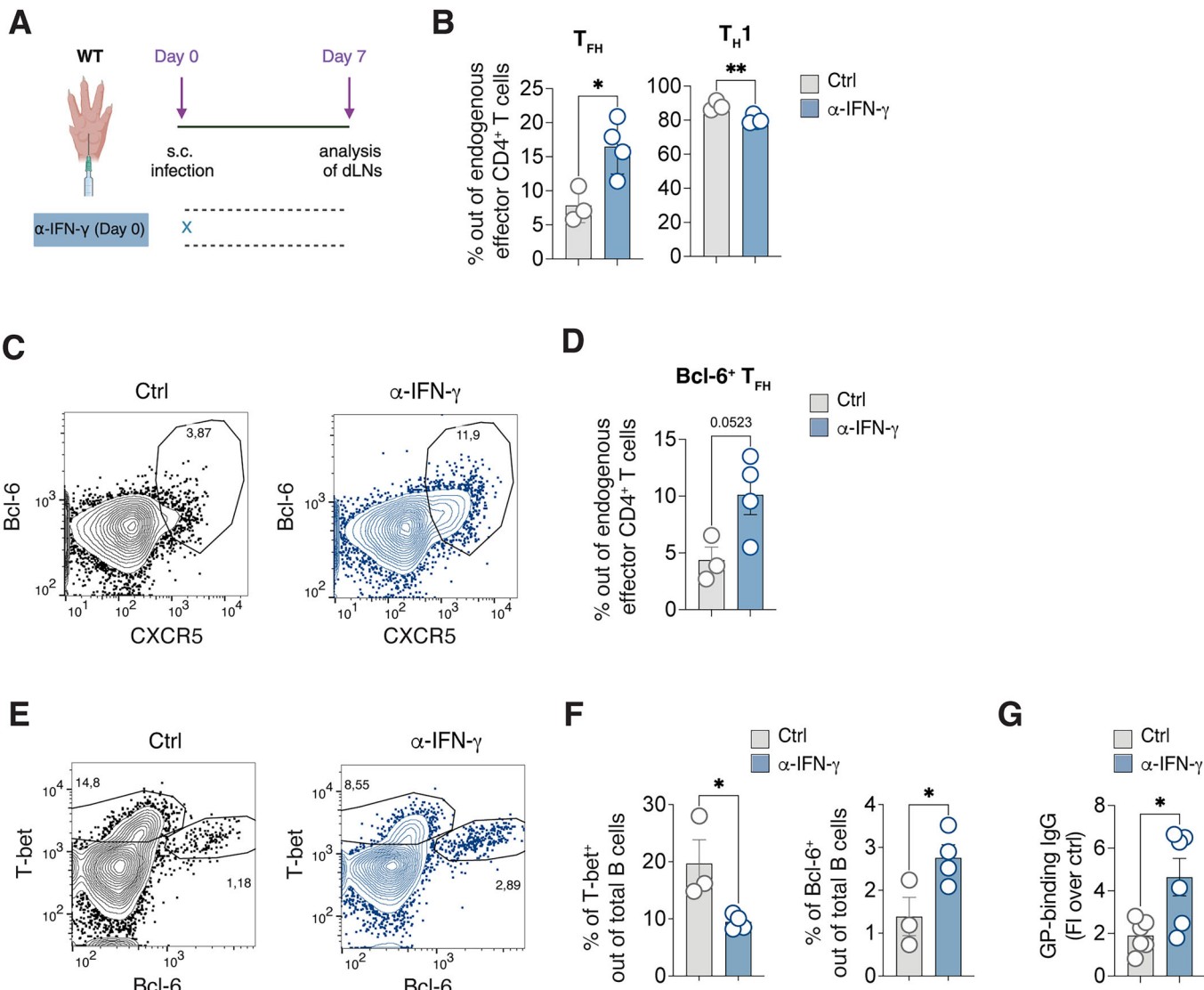

**Figure 4. IFN-γ suppresses T follicular helper cell and B cell responses in an endogenous setting.**

(A) CD45.2⁺ WT mice were infected s.c. with rLCMV (1*10⁵ FFU /footpad) and treated with α-IFN-γ blocking antibody (or isotype Ctrl) at day 0. dLNs were analyzed 7 days post infection. (B) Quantification of T_FH (left) and T_H1 (right), expressed as percentages out of endogenous effector CD4⁺ T cells in dLNs of s.c. infected mice described in (A). $n = 3$–4. Mean ± SEM is shown. Data are representative of three independent experiments. An unpaired two-tailed t test was applied. *$p$-value = 0.0231, **$p$-value = 0.0085. (C) Representative flow cytometry plots showing CXCR5⁺Bcl-6⁺ cells expressed as percentages among endogenous effector CD4⁺ T cells in the dLNs. Numbers represent the percentage of cells within the indicated gate. (D) Quantification of Bcl-6⁺ T_FH, expressed as percentages out of endogenous effector CD4⁺ T cells in dLNs of s.c. infected mice described in (A). $n = 3$–4. Mean ± SEM is shown. Data are representative of three independent experiments. An unpaired two-tailed t test was applied. $p$-value = 0.0523. (E) Representative flow cytometry plots showing T-bet⁺ and Bcl-6⁺ B cells expressed as percentages among total B cells (B220⁺) in the dLNs. Numbers represent the percentage of cells within the indicated gate. (F) Quantification of T-bet⁺ cells (left) and Bcl-6⁺ cells (right) expressed as percentages out of total B cells in dLNs of mice described in (A). $n = 3$ (Ctrl), 4 (α-IFN-γ). Mean ± SEM is shown. Data are representative of three independent experiments. An unpaired two-tailed t test was applied. *$p$-value = 0.0360 (T-bet⁺) and 0.0457 (Bcl-6⁺). (G) LCMV GP-binding IgG Abs were measured in the sera of mice 7 days upon infection as described in (A) and expressed as fold induction over uninfected controls. $n = 6$. Mean ± SEM is shown. Data were pooled from two independent experiments. An unpaired two-tailed t test was applied. *$p$-value = 0.0148. Source data are available online for this figure.

leaving only IFN-γ⁻/⁻ DCs for T cell priming, did not impair CD4⁺ T cell differentiation (Appendix Fig. S11A,B). Similarly, the lack of monocytes in CCR2-deficient mice did not significantly impact CD4⁺ T cell polarization post-infection (Appendix Fig. S12A–C).

Subsequently, we focused on T cells, as they were identified as early IFN-γ producers (Fig. 6A and Fig. 5G). Depletion of CD8⁺ T cells slightly increased T_FH and Tcf-1⁺ subsets while reducing

T_H1 and GzmB⁺ subsets (Fig. 6D and Appendix Fig. S13A,B). A more pronounced effect was observed when combining CD8⁺ T cell depletion with IFN-γ blockade, suggesting that CD8⁺ T cells likely contribute to CD4⁺ T cell polarization through mechanisms other than IFN-γ (Fig. 6D; Appendix Fig. S13A,B).

Lastly, the role of CD4⁺ T cell-derived IFN-γ was examined. We transferred Smarta WT or IFN-γ⁻/⁻ CD4⁺ T cells into WT or IFN-γ⁻/⁻

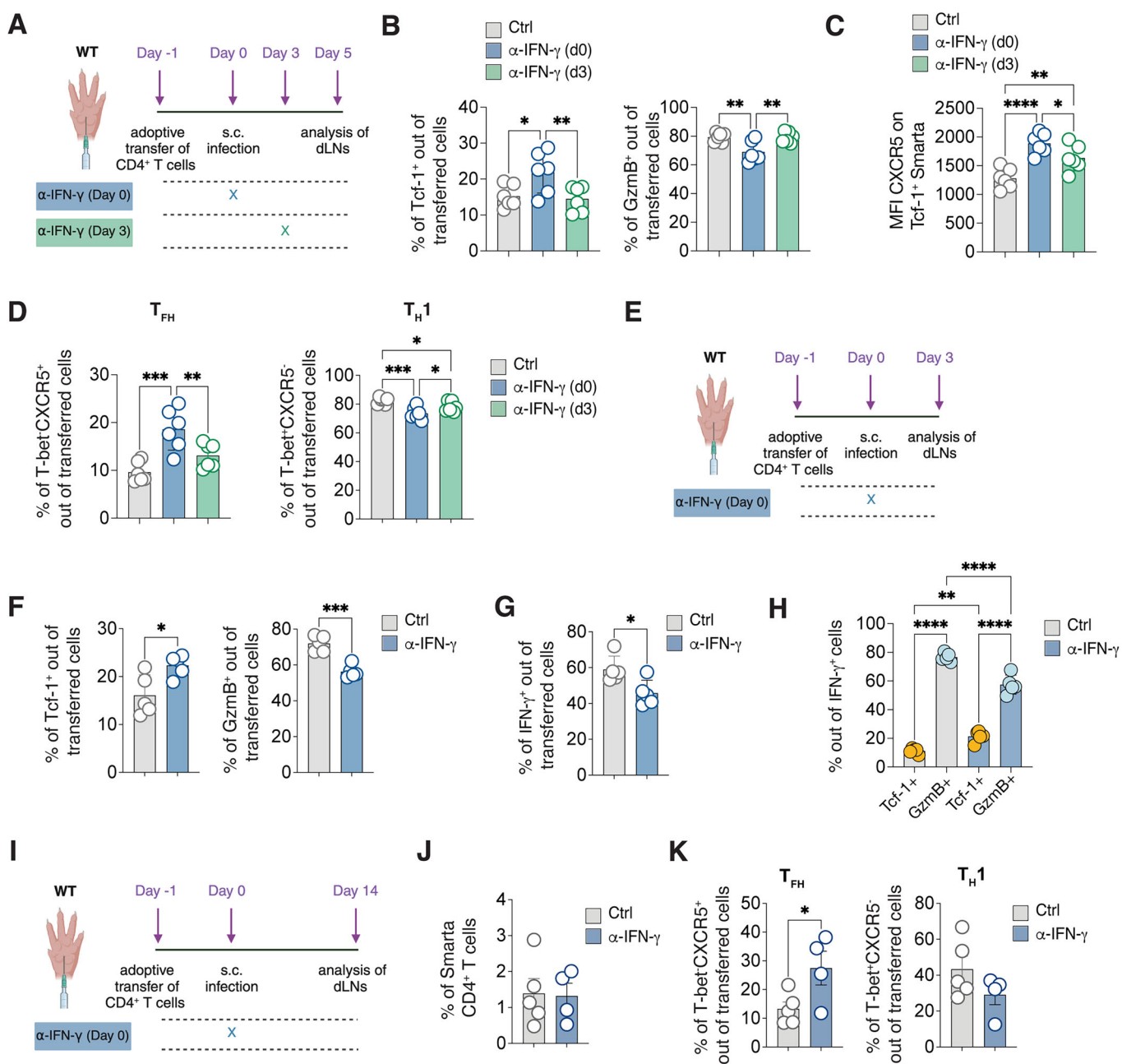

recipients and assessed Smarta polarization post-infection. Consistent with earlier findings, the absence of IFN-γ in both donor and recipient led to an increase in T_FH cells and a marked decrease in T_H1 cells (Fig. 6E). Smarta-derived IFN-γ effectively suppressed T_FH polarization in IFN-γ^{-/-} recipients, almost as efficiently as in WT recipients. However, when Smarta cells could not produce IFN-γ, a majority of CD4^+ T cells still differentiated into T_H1, implying that IFN-γ from endogenous CD4^+ and CD8^+ T cells might suffice for T_FH suppression (Fig. 6E; Appendix Fig. S14). These results collectively suggest that IFN-γ from adoptively transferred CD4^+ T cells, as well as endogenous T cells, is responsible for T_FH suppression in the context of s.c. LCMV infection.

## IFN-γ produced by T cells does not act in an autocrine fashion

Other groups have previously shown that IFN-γ produced by T_H1 cells plays an instrumental role in maintaining and reinforcing the T_H1 phenotype by acting in an autocrine fashion (Bradley et al, 1996; Zhang et al, 2001). To investigate whether early IFN-γ needed to suppress T_FH differentiation in the s.c. LCMV infection also targets CD4^+ T cells, we applied Crispr/Cas9 technology to delete IFNGR1 from primary Smarta CD4^+ T cells (Fig. 7A). Three control sgRNAs or three sgRNAs targeting IFNGR1 were transfected into naive Smarta cells with high efficiency (Fig. 7B)

**Figure 5. The IFN-γ responsible for T_FH suppression is produced in the first days upon infection.**

(A) $0.5*10^6$ purified CD45.1⁺ Smarta CD4⁺ T cells were transferred into CD45.2⁺ WT recipients 1 day before s.c. rLCMV infection ($1*10^5$ FFU /footpad). CD45.2⁺ WT recipient mice were also treated with α-IFN-γ blocking antibody (or isotype Ctrl) at day 0 or d3 after infection. dLNs were analyzed 5 days post infection. (B) Quantification of Tcf-1⁺ (left) and GzmB⁺ (right), expressed as percentages out of transferred Smarta CD4⁺ T in dLNs of mice described in (A). $n = 6$. Mean ± SEM is shown. Data are representative of three independent experiments. One-way ANOVA with uncorrected Fisher's LSD was applied. *p-value = 0.0175, **p-value = 0.0097 (Tcf-1⁺), **p-value = 0.0029 (GzmB⁺, Ctrl vs αIFN-γ d0), **p-value = 0.0037 (GzmB⁺, αIFN-γ d0 vs αIFN-γ d3). (C) Quantification of the MFI of CXCR5 on Tcf-1⁺ Smarta CD4⁺ T cells in dLNs of mice described in (A). $n = 6$. Mean ± SEM is shown. Data are representative of three independent experiments. One-way ANOVA with uncorrected Fisher's LSD was applied. *p-value = 0.0411 (αIFN-γ d0 vs αIFN-γ d3), **p-value = 0.007 (Ctrl vs αIFN-γ d3), ****p-value = 0.00008 (Ctrl vs αIFN-γ d0). (D) Quantification of T_FH (left) and T_H1 (right), expressed as percentages out of transferred Smarta CD4⁺ T cells in dLNs of mice described in (A). $n = 6$. Mean ± SEM is shown. Data are representative of three independent experiments. One-way ANOVA with uncorrected Fisher's LSD was applied. **p-value = 0.0088, ***p-value = 0.0002 (T_FH), *p-value = 0.0420 (T_H1, αIFN-γ d0 vs αIFN-γ d3), *p-value = 0.0407 (T_H1, Ctrl vs αIFN-γ d3), ***p-value = 0.0005 (T_H1, Ctrl vs αIFN-γ d0). (E) $0.5*10^6$ purified CD45.1⁺ Smarta CD4⁺ T cells were transferred into CD45.2⁺ WT recipients 1 day before s.c. rLCMV infection ($1*10^5$ FFU /footpad). CD45.2⁺ WT recipient mice were also treated with α-IFN-γ blocking antibody (or isotype Ctrl) at day 0. dLNs were analyzed 3 days post infection. (F) Quantification of Tcf-1⁺ and GzmB⁺ cells, expressed as percentages out of transferred Smarta CD4⁺ T cells in dLNs of mice described in (E). $n = 5$. Mean ± SEM is shown. Data are representative of two independent experiments. An unpaired two-tailed t test was applied. *p-value = 0.0201, ***p-value = 0.0002. (G) Quantification of IFN-γ⁺ cells out of ex-vivo restimulated Smarta CD4⁺ T cells in dLNs of mice described in (E). $n = 5$. Mean ± SEM is shown. Data are representative of two independent experiments. An unpaired two-tailed t test was applied. *p-value = 0.0222. (H) Quantification of GzmB⁺ and Tcf-1⁺ cells expressed as percentages among IFN-γ⁺ Smarta CD4⁺ T cells in dLNs of mice described in (E). $n = 5$. Mean ± SEM is shown. Data are representative of two independent experiments. One-way ANOVA with uncorrected Fisher's LSD was applied. **p-value = 0.001462391 (Tcf-1⁺ Ctrl vs Tcf-1⁺ αIFN-γ), ****p-value < 0.000000001 (Tcf-1⁺ Ctrl vs GzmB⁺ Ctrl), ****p-value = 0.000001960 (GzmB⁺ Ctrl vs GzmB⁺ αIFN-γ), ****p-value < 0.000000001 (Tcf-1⁺ αIFN-γ vs GzmB⁺ αIFN-γ). (I) $0.5*10^6$ purified CD45.1⁺ Smarta CD4⁺ T cells were transferred into CD45.2⁺ WT recipients 1 day before s.c. rLCMV infection ($1*10^5$ FFU /footpad). CD45.2⁺ WT recipient mice were also treated with α-IFN-γ blocking antibody (or isotype Ctrl) at day 0. dLNs were analyzed 14 days post infection. (J) Quantification of Smarta CD4⁺ T cells, expressed as percentages out of total LNs cells of mice described in (D). (K) Quantification of T_FH (left) and T_H1 (right), expressed as percentages out of transferred Smarta CD4⁺ T cells in dLNs of mice described in (D). $n = 5$ (Ctrl), 4 (α-IFN-γ). Mean ± SEM is shown. Data are representative of two independent experiments. An unpaired two-tailed t test was applied. *p-value = 0.0455.

prior to their transfer into WT recipient mice. Five days upon s.c. LCMV infection we found that Smarta cells had expanded similarly in the two groups of recipients (Fig. 7C) and Smarta cells receiving sgRNAs targeted to IFNGR1 displayed lower levels of the IFNGR1 protein (Fig. 7D). Notably, no difference at all was observed in CD4⁺ T cell polarization (Fig. 7E). To exclude any technical caveats we adopted an alternative approach by transferring into WT recipients genetically modified Smarta IFNGR1⁻/⁻ or Smarta WT mice (Fig. 7F). In addition, both experimental groups were treated or not with the IFN-γ blocking Ab. In this setting we detected significantly lower levels of IFNGR1 protein on Smarta IFNGR1⁻/⁻, as well as slightly lower levels of the receptor on WT Smarta in mice treated with the IFN-γ blocking Ab, suggesting that CD4⁺ T cells can upregulate the receptor when they sense IFN-γ (Fig. 7G). However, no significant differences were observed in the polarization between WT and IFNGR1⁻/⁻ cells (Fig. 7H). Instead, a notable T_FH rescue (expressing both CXCR5 and Bcl-6) was observed when mice were treated with the IFN-γ blocking Ab (Fig. 7H). All in all, these experiments show that IFN-γ released by T cells suppresses T_FH differentiation by acting on cells other than CD4⁺ T cells.

To determine which cells of the LN might be the target for IFN-γ, we analyzed IFNGR1 protein levels on different immune cells in the first three days upon s.c. LCMV infection (Fig. 7I). As suggested by abovementioned results, CD4⁺ T cells did not express high levels of the receptor, and the same was for CD8⁺ T cells, B cells, and granulocytes. Instead, the highest levels of IFNGR1 were expressed by group 1 ILCs and DC (Fig. 7I). Since depletion of group 1 ILCs did not have any effect on CD4⁺ T cell polarization (Fig. 6B,C), we hypothesized that the possible target for IFN-γ suppressing T_FH differentiation might be DC. Analysis of a previous published dataset (Data ref: De Giovanni et al, 2020) confirmed that DC sense IFN-γ during LCMV infection and express a signature of IFN-γ-stimulated genes 48 h after infection (Fig. 7J; Appendix Fig. S15A). In addition, analysis of a published transcriptional dataset on WT and IFNGR2⁻/⁻ DCs during a parasitic infection (Data ref: Lee et al, 2015a, 2015b) confirmed that DC that

sense IFN-γ are characterized by a different transcriptional profile with respect to those that cannot sense this cytokine (Appendix Fig. S15B). In summary, our findings indicate that the IFN-γ responsible for T_FH suppression is T cell-derived but does not act on CD4⁺ T cells themselves. Instead, IFN-γ might be sensed by another cell type, possibly DCs, which might then acquire a phenotype that interferes with T_FH differentiation. However, further functional studies are required to corroborate this hypothesis.

## Exploring the role of IFN-γ across various infection and immunization models

Diverging from s.c. infection dynamics, systemic LCMV infection notably induces a dual polarization of CD4⁺ T cells into both T_H1 and T_FH subsets (Johnston et al, 2009; Hale et al, 2013; Ray et al, 2014; Weinstein et al, 2018). Echoing previous studies, we observed that approximately 40–50% of Smarta CD4⁺ T cells adopted a CXCR5⁺ profile indicative of T_FH cells on day 5 post systemic LCMV infection. The remaining population predominantly exhibited heightened T-bet expression (Appendix Fig. S16A,B). In this context, IFN-γ blockade distinctly influenced CD4⁺ T cell polarization, slightly increasing T_FH frequencies while decreasing T_H1 cell proportions (Fig. 8A,B). However, this modulation only minimally altered Tcf-1⁺ and GzmB⁺ subsets, nor did it enhance CXCR5 expression within the Tcf-1⁺ subset, contrasting the patterns observed in s.c. infection routes (Fig. 2B,C; Appendix Fig. S16C–E).

We hypothesized that these observed differences might stem from varying IFN-γ levels induced by different infection routes. Indeed, quantitative PCR analysis revealed that systemic infection elicited lower Ifng levels compared to s.c. infection (Fig. 8C). Considering potential overestimation of Ifng expression due to the adoptive transfer of antigen-specific CD4⁺ T cells, we assessed endogenous Ifng expression in non-transferred mice. This confirmed significant Ifng induction only in s.c.ly infected mice (Fig. 8D). This finding aligns with

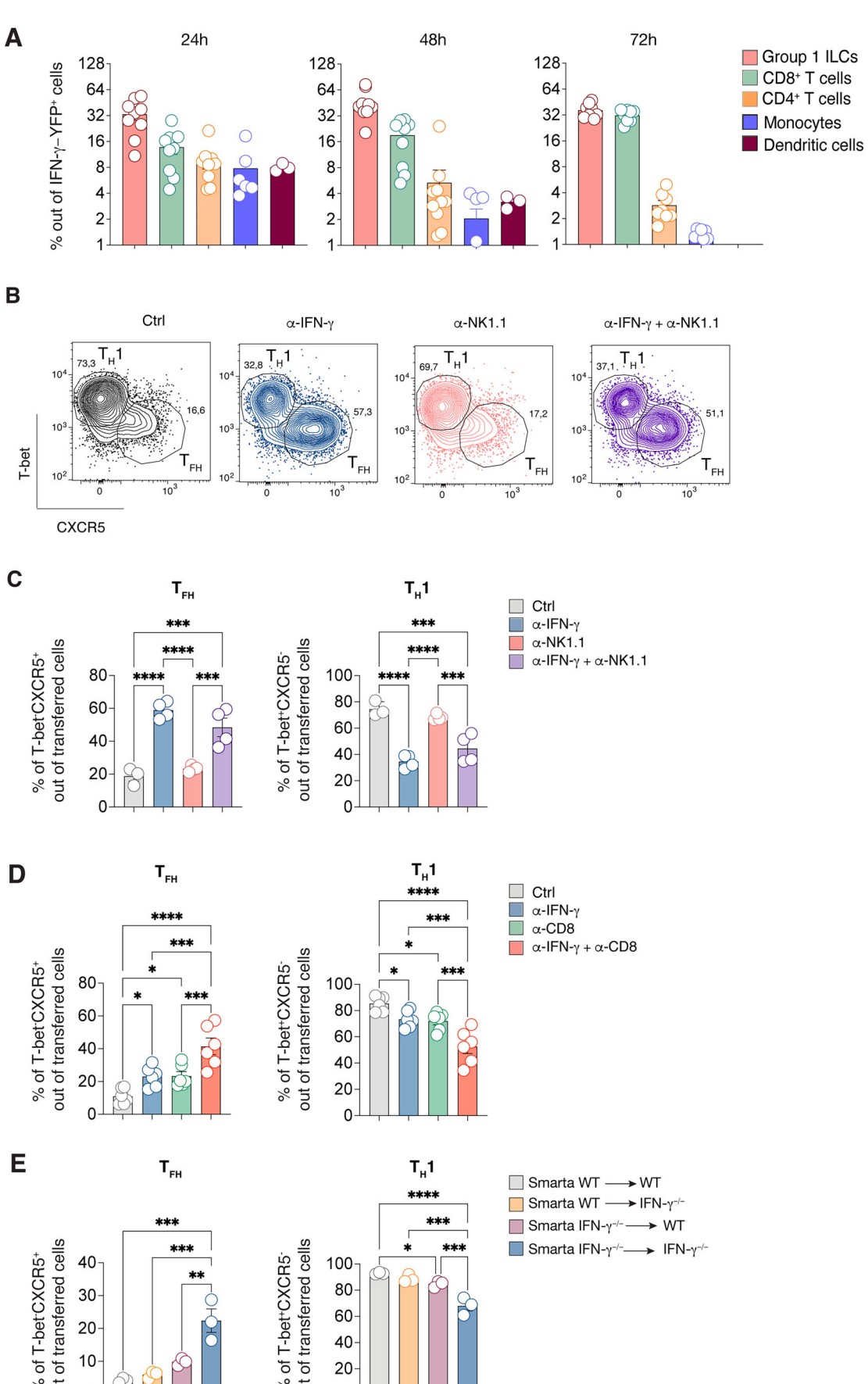

**Figure 6. IFN-γ produced by T cells is responsible for T$_{FH}$ suppression.**

(A) dLNs of IFN-γ-YFP mice were analyzed at 24 h, 48 h and 72 h upon s.c. rLCMV infection (1*10$^5$ FFU/footpad). Quantification of Group 1 ILCs (NK1.1$^+$ NKp46$^+$), CD8$^+$, CD4$^+$, monocytes (CD11b$^+$ Ly6C$^{hi}$), DC (CD11c$^+$ MHC-II$^+$) expressed as percentages out of the total YFP$^+$ cells is shown. $n$ = 3–9 (24, 48 h), 7 (72 h). Mean ± SEM is shown. Data are representative of two independent experiments. (B) 0.5*10$^6$ purified CD45.1$^+$ Smarta CD4$^+$ T cells were transferred into CD45.2$^+$ WT recipients 1 day before s.c. rLCMV infection (1*10$^5$ FFU/footpad). In some conditions CD45.2$^+$ WT recipient mice were also treated with α-IFN-γ blocking antibody at day 0, α-NK1.1 antibody (d-1, d0) or both antibodies in combination. dLNs were analyzed 5 days post infection. Representative flow cytometry plot showing T$_{FH}$ and T$_H$1 cells among Smarta CD4$^+$ T cells in dLNs. Numbers represent the percentage of cells within the indicated gate. (C) Quantification of T$_{FH}$ and T$_H$1 cells, expressed as percentages out of transferred Smarta CD4 + T cells, in dLNs of mice described in (B). $n$ = 3 (Ctrl), 4 (α-IFN-γ, α-NK1.1, α-IFN-γ + α-NK1.1). Mean ± SEM is shown. Data are representative of three independent experiments. One-way ANOVA with uncorrected Fisher's LSD was applied. ****$p$-value = 0.000010120 (T$_{FH}$, Ctrl vs αIFN-γ), ***$p$-value = 0.000312 (T$_{FH}$, αNK1.1 vs αNK1.1 + αIFN-γ), ***$p$-value = 0.00015 (T$_{FH}$, Ctrl vs αNK1.1 + αIFN-γ), ****$p$-value = 0.000014996 (T$_{FH}$, αIFN-γ vs αNK1.1), ****$p$-value = 0.000009250 (T$_H$1, Ctrl vs αIFN-γ), ***$p$-value = 0.000426 (T$_H$1, αNK1.1 vs αNK1.1 + αIFN-γ), ***$p$-value = 0.00012 (T$_H$1, Ctrl vs αNK1.1 + αIFN-γ), ****$p$-value = 0.000021676 (T$_H$1, αIFN-γ vs αNK1.1). (D) 0.5*10$^6$ purified CD45.1$^+$ Smarta CD4$^+$ T cells were transferred into CD45.2$^+$ WT recipients 1 day before s.c. rLCMV infection (1*10$^5$ FFU /footpad). In some conditions CD45.2$^+$ WT recipient mice were also treated with α-IFN-γ blocking antibody at day 0, α-CD8 antibody (d-1, d2) or both antibodies in combination. dLNs were analyzed 5 days post infection. Quantification of T$_{FH}$ and T$_H$1 cells, expressed as percentages out of transferred Smarta CD4$^+$ T cells. $n$ = 6. Mean ± SEM is shown. Data are representative of two independent experiments. One-way ANOVA with uncorrected Fisher's LSD was applied. *$p$-value = 0.0168 (T$_{FH}$, Ctrl vs αIFN-γ), *$p$-value = 0.0146 (T$_{FH}$, Ctrl vs αCD8), ****$p$-value = 0.000002018 (T$_{FH}$, Ctrl vs αCD8 + αIFN-γ), ***$p$-value = 0.00073 (T$_{FH}$, αIFN-γ vs αCD8 + αIFN-γ), ***$p$-value = 0.00084 (T$_{FH}$, αCD8 vs αCD8 + αIFN-γ), *$p$-value = 0.0233 (T$_H$1, Ctrl vs αIFN-γ), *$p$-value = 0.0129 (T$_H$1, Ctrl vs αCD8), ****$p$-value = 0.000001653 (T$_H$1, Ctrl vs αCD8 + αIFN-γ), ***$p$-value = 0.00041 (T$_H$1, αIFN-γ vs αCD8 + αIFN-γ), ***$p$-value = 0.00077 (T$_H$1, αCD8 vs αCD8 + αIFN-γ). (E) 0.5*10$^6$ purified CD45.1$^+$ Smarta CD4$^+$ T cells from WT or Smarta-IFN-γ$^{-/-}$ were transferred into CD45.2$^+$ WT or IFN-γ$^{-/-}$ recipients 1 day before s.c. rLCMV infection (1*10$^5$ FFU /footpad). dLNs were analyzed 5 days post infection. Quantification of T$_{FH}$ and T$_H$1 cells, expressed as percentages out of transferred Smarta CD4$^+$ T cells. $n$ = 3. Mean ± SEM is shown. Data are representative of three independent experiments. One-way ANOVA with uncorrected Fisher's LSD was applied. ***$p$-value = 0.0001 (T$_{FH}$, Smarta WT in WT vs Smarta IFN-γ$^{-/-}$ in IFN-γ$^{-/-}$), ***$p$-value = 0.0002 (T$_{FH}$, Smarta WT in IFN-γ$^{-/-}$ vs Smarta IFN-γ$^{-/-}$ in IFN-γ$^{-/-}$), **$p$-value = 0.0013 (T$_{FH}$, Smarta IFN-γ $^{-/-}$ in WT vs Smarta IFN-γ$^{-/-}$ in IFN-γ$^{-/-}$), ****$p$-value = 0.000045689 (T$_H$1, Smarta WT in WT vs Smarta IFN-γ$^{-/-}$ in IFN-γ$^{-/-}$), ***$p$-value = 0.00016 (T$_H$1, Smarta WT in IFN-γ$^{-/-}$ vs Smarta IFN-γ$^{-/-}$ in IFN-γ$^{-/-}$), ***$p$-value = 0.00068 (T$_H$1, Smarta IFN-γ$^{-/-}$ in WT vs Smarta IFN-γ$^{-/-}$ in IFN-γ$^{-/-}$), *$p$-value = 0.032 (T$_H$1, Smarta WT in WT vs Smarta IFN-γ$^{-/-}$ in WT).

previous reports suggesting that systemic LCMV infection's low *Ifng* expression could be a result of type I IFNs inhibition (Cousens et al, 1997b; Nguyen et al, 2000; Pien and Biron, 2000; Nguyen et al, 2002; Miyagi et al, 2007). Consistently, endogenous CD4$^+$ T cell polarization post systemic infection remained unchanged despite IFN-γ blockade (Appendix Fig. S16F–H), whereas as shown before s.c. infection saw a notable rescue of T$_{FH}$ cells (Fig. 4).

To probe for a possible role of IFN-γ in affecting CD4 T cells responses in other infection settings, we decided to analyze CD4$^+$ T cell polarization in the context of a mouse model for SARS-CoV-2 infection. To this end we used a mouse-adapted SARS-CoV-2 strain, rSARS-N501YMA30 (Wong et al, 2022) which was administered to WT mice via aerosol (Fumagalli et al, 2021, 2024) (Fig. 8E). We found that SARS-CoV-2 infection induced a balanced CD4$^+$ T cell response consisting of both T$_H$1 and T$_{FH}$ cells: however, when blocking IFN-γ, T$_H$1 cells decreased significantly, whereas T$_{FH}$ frequency increased (Fig. 8F), confirming the role of IFN-γ as a molecular switch shifting the balance towards cellular responses. Moreover, two weeks after infection, we observed a trend in the increase of IgG1 and IgG3 antibodies, which have been reported to be the only T$_{FH}$-dependent isotypes (Chen et al, 2022) (Fig. 8G).

Broadening our investigation, we explored the role of IFN-γ in a translational context using a monophosphoryl lipid A (MPLA)-based immunization model known to foster T$_H$1 differentiation (Mata-Haro et al, 2007; Komai-Koma et al, 2021). Mice were immunized with MPLA-conjugated SARS-CoV-2 RBD protein, and the effects of IFN-γ blockade were analyzed (Fig. 8H). Immunization resulted in increased expression of *Ifng* especially at 16 h (Appendix Fig. S16I). Intriguingly, inhibiting IFN-γ in this setting also led to reduced T$_H$1 cells, and increased T$_{FH}$ and Bcl-6$^+$ B cells (Fig. 8I,J). These findings collectively suggest that both in viral infection and immunization scenarios, IFN-γ functions to suppress T$_{FH}$ responses, and its blockade could enhance humoral vaccine responses.

# Discussion

CD4$^+$ T cell polarization plays a critical role in shaping effector immune responses (Tuzlak et al, 2021). Previously, we demonstrated that s.c. LCMV infection is characterized by a stark compartmentalization of CD4$^+$ T cell responses, leading to almost exclusive T$_H$1 polarization but severely impaired T$_{FH}$ cell differentiation (De Giovanni et al, 2020). Building upon these findings, our current research reveals that IFN-γ, known for supporting type I responses and maintaining the T$_H$1 phenotype, plays a pivotal role in suppressing T$_{FH}$ differentiation and B cell responses during viral infections. Importantly, we found that the IFN-γ responsible for T$_{FH}$ suppression is T cell-derived but does not act on CD4$^+$ T cells themselves, unlike to what previously reported for T cell expansion and T$_H$1 maintenance (Bradley et al, 1996; Lighvani et al, 2001; Whitmire et al, 2005). Instead, IFN-γ might be sensed by DCs, which might then acquire a phenotype that interferes with T$_{FH}$ differentiation. Although the exact cell type targeted by IFN-γ should be further corroborated by functional studies, we propose that T$_H$1 antagonize T$_{FH}$ not only through competition of cell-intrinsic transcription factors as proposed by others (Nakayamada et al, 2011), but also through a cell-extrinsic effect on the surrounding microenvironment.

The inhibitory effect of IFN-γ on T$_{FH}$ differentiation and humoral immunity is not unprecedented. For instance, in severe malaria infection, a setting where T$_H$1-like dysfunctional T$_{FH}$ cells expressing T-bet are observed, concomitant blockade of IFN-γ and TNF-α resulted in enhanced T$_{FH}$ differentiation and improved antibody responses (Ryg-Cornejo et al, 2016). Nevertheless, in this context, the cells responsible for producing or sensing IFN-γ were not identified, and the underlying mechanism remained unexplored. In another study performed with bacterial infection, IFN-γ produced by B cells upon IL-12 sensing was shown to act in an autocrine fashion and to suppress GC reactions (Elsner et al, 2024).

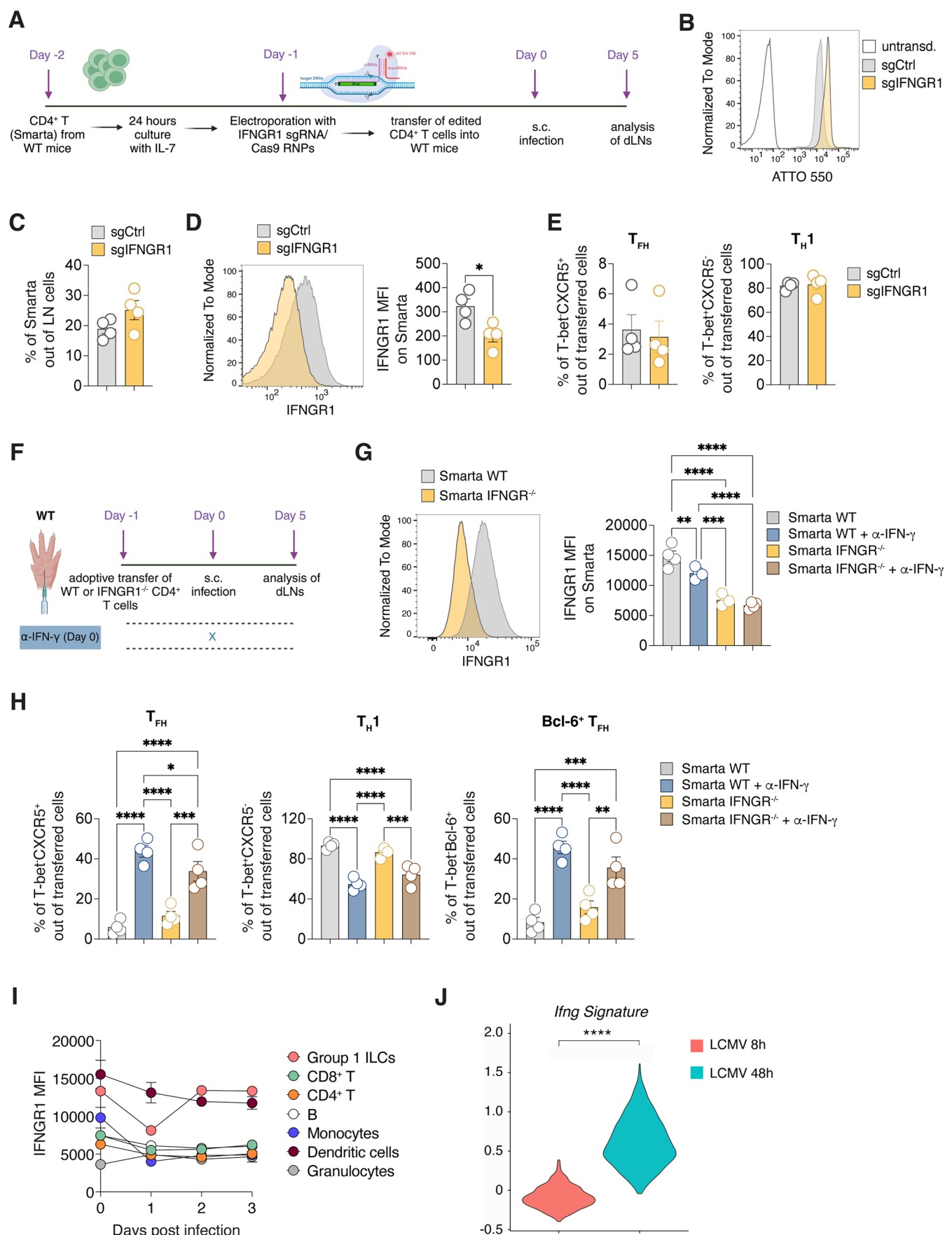

◀ **Figure 7. IFN-γ produced by T cells does not act in an autocrine fashion.**

(A) Total CD4[+] T cells were isolated from spleens of naive CD45.1[+] Smarta Tg mice and cultured for 24 h with rIL-7. The day after, Smarta CD4[+] T cells were transduced with control sgRNAs or with sgRNAs targeting IFNGR1 and with Cas9. After transduction, 0.5*10[6] purified CD45.1[+] Smarta CD4[+] T cells were transferred into CD45.2[+] WT 1 day before s.c. rLCMV infection (1*10[5] FFU /footpad). dLNs were analyzed 5 days post infection. (B) Representative plot of the transduction efficiency of Smarta CD4[+] T cells is shown. (C) Quantification of transferred Smarta CD4[+] T cells expressed as percentages. $n = 4$. Mean ± SEM is shown. Data are representative of two independent experiments. (D) Representative flow cytometry plot (left panel) and quantification (right panel) showing the fluorescence intensity of IFNGR1 expression within Smarta CD4[+] T cells of mice described in (A). $n = 4$. Mean ± SEM is shown. Data are representative of two independent experiments. An unpaired two-tailed t test was applied: $*p$-value = 0.0225. (E) Quantification of $T_{FH}$ (left) and $T_H1$ (right), expressed as percentages out of transferred Smarta CD4[+] T cells in dLNs of mice described in (A). $n = 4$. Mean ± SEM is shown. Data are representative of two independent experiments. (F) 0.5*10[6] purified CD45.1[+] WT or Thy1.1[+] IFNGR1[−/−] Smarta CD4[+] T cells were transferred into CD45.2[+] WT 1 day before s.c. rLCMV infection (1*10[5] FFU /footpad). CD45.2[+] WT recipient mice were also treated with α-IFN-γ blocking antibody (or isotype Ctrl) at day 0. dLNs were analyzed 5 days post infection. (G) Representative flow cytometry plot (left panel) and quantification (right panel) showing the fluorescence intensity of IFNGR1 expression within Smarta CD4[+] T cells of mice described in (F). $n = 4$. Mean ± SEM is shown. Data are representative of three independent experiments. An unpaired two-tailed t test was applied. $**p$-value = 0.004408606 (Smarta WT vs Smarta WT + αIFN-γ), $****p$-value = 0.000001051 (Smarta WT vs Smarta KO), $****p$-value = 0.000000303 (Smarta WT vs Smarta KO + αIFN-γ), $***p$-value = 0.000126732 (Smarta WT + αIFN-γ vs Smarta KO), $****p$-value = 0.000023247 (Smarta WT + αIFN-γ vs Smarta KO + αIFN-γ). (H) Quantification of $T_{FH}$ (left), $T_H1$ (middle), and Bcl-6[+] $T_{FH}$ (right) expressed as percentages out of transferred Smarta CD4[+] T cells in dLNs of mice described in (F). $n = 4$. Mean ± SEM is shown. Data are representative of three independent experiments. $****p$-value = 0.00001977 ($T_{FH}$, Smarta WT vs Smarta WT + αIFN-γ), $****p$-value = 0.000040512 ($T_{FH}$, Smarta WT vs Smarta KO + αIFN-γ), $****p$-value = 0.000010401 ($T_{FH}$, Smarta WT + αIFN-γ vs Smarta KO), $*p$-value = 0.0455 ($T_{FH}$, Smarta WT + αIFN-γ vs Smarta KO + αIFN-γ), $***$ p-value = 0.000308926 ($T_{FH}$, Smarta KO vs Smarta KO + αIFN-γ); $****p$-value = 0.000002129 ($T_H1$, Smarta WT vs Smarta WT + αIFN-γ), $****p$-value = 0.000034278 ($T_H1$, Smarta WT vs Smarta KO + αIFN-γ), $****p$-value = 0.000014786 ($T_H1$, Smarta WT + αIFN-γ vs Smarta KO), $***p$-value = 0.000353765 ($T_H1$, Smarta KO vs Smarta KO + αIFN-γ); $****$ p-value = 0.000011768 (Bcl-6[+] $T_{FH}$, Smarta WT vs Smarta WT + αIFN-γ), $***p$-value = 0.000216252 (Bcl-6[+] $T_{FH}$, Smarta WT vs Smarta KO + αIFN-γ), $****p$-value = 0.000097437 (Bcl-6[+] $T_{FH}$, Smarta WT + αIFN-γ vs Smarta KO), $**p$-value = 0.002602899 (Bcl-6[+] $T_{FH}$, Smarta KO vs Smarta KO + αIFN-γ); (I) Quantification of the mean fluorescence intensity of IFNGR1 expression on the indicated immune cells subsets in dLNs at the indicated time-points upon s.c. LCMV infection. $n = 3$. Mean ± SEM is shown. (J) Violin plot representation of the enrichment scores of the *Ifng* signature, comparing DCs sorted from mice infected with LCMV for 8 or 48 h (published dataset in (Data ref: De Giovanni et al, 2020). Two-tailed Mann-Whitney test has been performed, $****p$-value < 0.0001. Source data are available online for this figure.

Our research indicates that, at least upon viral infection, IFN-γ specifically produced by T cells suppresses functional $T_{FH}$ differentiation by likely acting on DC. Since IFN-γ was found to be produced in the first three days upon infection, we hypothesize that the first differentiated $T_H1$ cells might antagonize the arising $T_{FH}$ cells by possibly modifying the dLNs microenvironment.

Notably, we found that IFN-γ does not influence only CD4[+] T cell polarization but also the B cell phenotype, as shown by restriction of the development of Bcl-6[+] B cells, commonly identified as GC B cells. A role for IFN-γ on B cell activation has been reported by others although with different outcomes depending on the context (Abed et al, 1994; Myles et al, 2017; Obeng-Adjei et al, 2017; Unger et al, 2018; Stone et al, 2019; Zumaquero et al, 2019; Chodisetti et al, 2020; Arroyo-Díaz et al, 2023). In particular previous literature has suggested that IFN-γ can act directly on B cells to induce T-bet expression (Obeng-Adjei et al, 2017; Stone et al, 2019; Zumaquero et al, 2019; Chodisetti et al, 2020). T-bet-expressing B cells, observed in various chronic infections, severe malaria, and autoimmune diseases, are often characterized as 'atypical' due to their dysfunctional traits and pathogenicity (Obeng-Adjei et al, 2017; Rubtsov et al, 2017; Rubtsova et al, 2017; Burton et al, 2018). However, in the context of respiratory viral infections, T-bet-expressing T and B lymphocytes function collaboratively, facilitating optimal antiviral immunity (Mendoza et al, 2021). In our experimental setting IFNGR1 expression on B cells was not very high in the first three days upon infection, therefore we believe it is likely that the increase in Bcl6[+] B cells may result from enhanced $T_{FH}$ differentiation following early IFN-γ blockade, whereas the decrease in T-bet[+] B cells could be due to IFN-γ action directly on B cells at later time-points.

Recently, a population of Tcf-1[+]Bcl-6[low]PD-1[+] CD4[+] T cells that can give rise to both effector cells and $T_{FH}$ cells has been described in a setting of chronic LCMV infection and antigen persistence (Xia et al, 2022). We cannot formally exclude that this population functionally resembles Tcf-1[+]T-bet[+] CD4[+] T identified in our setting: however, it is worth mentioning that the s.c. LCMV

infection is cleared within one week and therefore represents an acute infection setting (Sammicheli et al, 2016).

Notably, IL-12, a well-established $T_H1$-polarizing cytokine (Hsieh et al, 1993; Heufler et al, 1996; Athie-Morales et al, 2004) was not essential for $T_H1$ differentiation in LCMV infection, as echoed by other studies (Schijns et al, 1998; Oxenius et al, 1999). This may be explained by the poor induction of IL-12 in certain viral infections due to type I IFNs' inhibitory effects (Cousens et al, 1997a; Pien and Biron, 2000). IFN-γ, while known to promote $T_H1$ phenotype survival and even act as a polarizing cytokine alongside IL-12 in some contexts (Bradley et al, 1996; Heufler et al, 1996; Wakil et al, 1998; Lighvani et al, 2001; Miro et al, 2006; Schulz et al, 2009), was crucial for the development of at least one identified $T_H1$ subset expressing both *Gzma* and *Gzmb*. The detailed characterization of this subset warrants further investigation.

Our study indicates that the pronounced CD4[+] T cell compartmentalization observed in s.c. LCMV infection, leading to a dominant $T_H1$ response, is likely attributed to the high levels of IFN-γ induced in this infection route. Indeed, we suggest that reduced IFN-γ induction in systemic infections (Cousens et al, 1997b; Nguyen et al, 2000; Pien and Biron, 2000; Nguyen et al, 2002; Miyagi et al, 2007) permits a coexistence of $T_H1$ and $T_{FH}$ cells (Johnston et al, 2009; Hale et al, 2013; Ray et al, 2014; Weinstein et al, 2018). In scenarios where IFN-γ is overexpressed, such as upon transfer of antigen-specific CD4[+] T cells, blocking IFN-γ results in increased $T_{FH}$ cell frequencies regardless of the infection route. However, IFN-γ was shown to restrict $T_{FH}$ development also in endogenous settings of not only s.c. LCMV infection, but also of SARS-CoV-2 infection or with an immunization approach. These findings imply a universal role for IFN-γ in suppressing humoral responses across various contexts. Crucially, this insight could guide the development of more effective vaccination strategies. Future studies delving deeper into the spatiotemporal mechanisms employed by this cytokine could provide further clarity and direction for innovative therapeutic approaches.

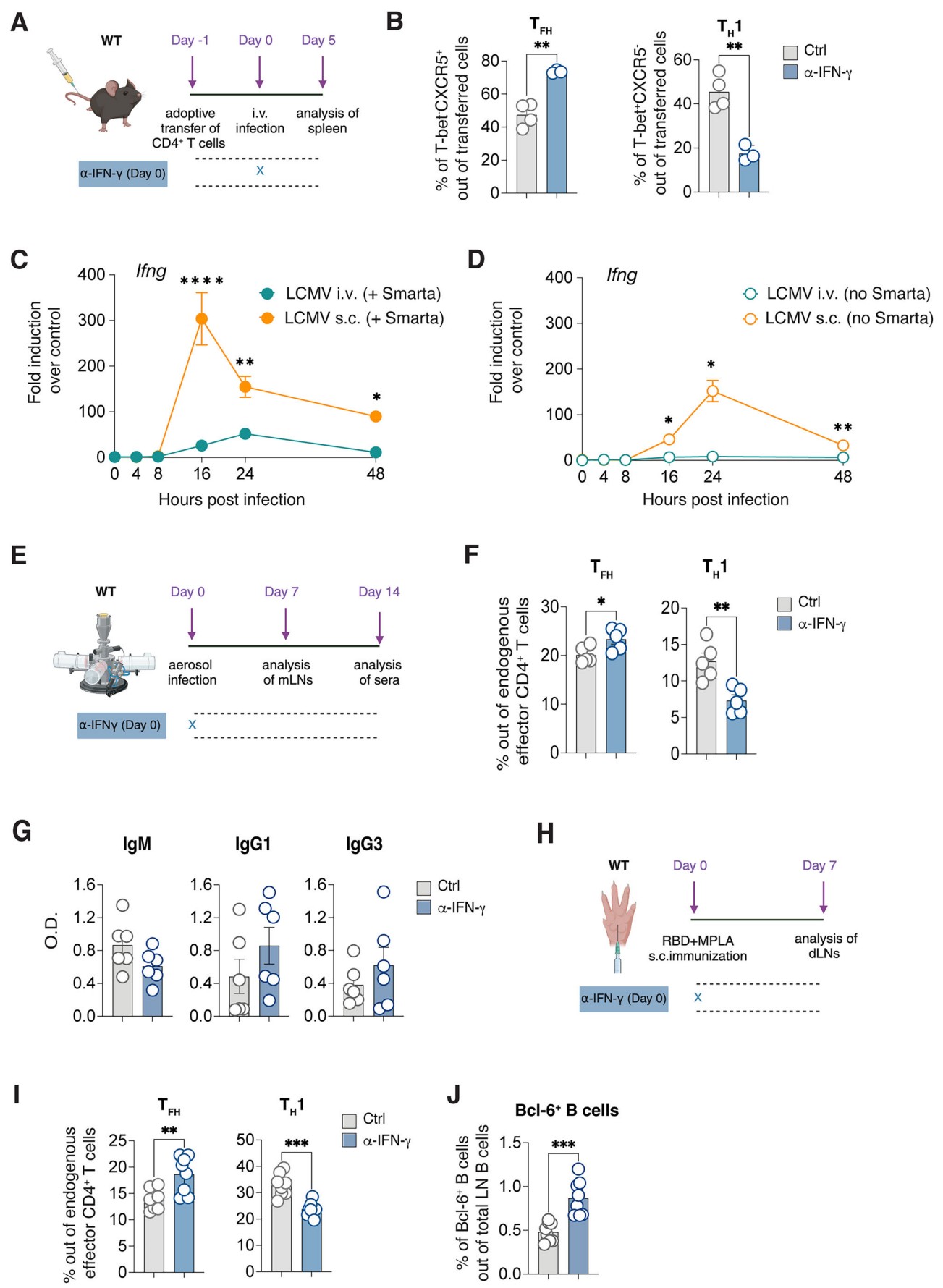

**Figure 8. Dissecting the role of IFN-γ in other infection and immunization settings.**

(A) 0.5*10⁶ purified CD45.1⁺ Smarta CD4⁺ T cells were transferred into CD45.2⁺ WT recipients 1 day before intravenous (i.v.) rLCMV infection (2*10⁵ FFU). CD45.2⁺ WT recipient mice were also treated with α-IFN-γ blocking antibody (or isotype Ctrl) at day 0. Spleens were analyzed 5 days post infection. (B) Quantification of T_FH (left) and T_H1 (right), expressed as percentages out of transferred Smarta CD4⁺ T cells in spleens of mice described in (A). *n* = 4 (Ctrl), 3 (α-IFN-γ). Mean ± SEM is shown. Data are representative of three independent experiments. An unpaired two-tailed t test was applied. **p-value = 0.0016 (T_FH) and 0.0023 (T_H1). (C) 0.5*10⁶ purified CD45.1⁺ Smarta CD4⁺ T cells were transferred into CD45.2⁺ WT recipients 1 day before i.v. (2*10⁵ FFU) or s.c. (1*10⁵ FFU /footpad) rLCMV infection. Analysis of *Ifng* gene expression at 0, 4, 8, 16, 24, and 48 h in dLN (orange) and spleens (green) of mice infected s.c. (orange) and i.v. (green) with rLCMV is shown. *n* = 4–6. Mean ± SEM is shown. Two-way ANOVA with uncorrected Fisher's LSD was applied. ****p-value < <0.000000001 (16 h), **p-value = 0.0052 (24 h), *p-value = 0.029 (48 h). (D) WT mice were infected i.v. (2*10⁵ FFU) or s.c. (1*10⁵ FFU /footpad) with rLCMV. Analysis of *Ifng* gene expression at 0, 4, 8, 16, 24 and 48 h in dLN (orange) and spleens (green) of mice infected s.c. (orange) and i.v. (green) with rLCMV is shown. *n* = 3–4. Mean ± SEM is shown. Two-way ANOVA with uncorrected Fisher's LSD was applied. *p-value = 0.042 (16 h), *p-value = 0.0249 (24 h), **p-value = 0.006 (48 h). (E) WT mice were infected via aerosol with a mouse-adapted strain of SARS-CoV-2 and mediastinal LNs (mLNs) were analyzed 7 days upon infection. (F) Quantification of T_FH (left) and T_H1 (right), expressed as percentages out of endogenous effector CD4⁺ T cells in mLNs of infected mice described in (G). *n* = 5. Mean ± SEM is shown. Data are representative of two independent experiments. An unpaired two-tailed t test was applied. *p-value = 0.0324 (T_FH), **p-value = 0.0049 (T_H1). (G) RBD–binding IgM, IgG1 and IgG3 Abs were measured in the sera of mice 14 days upon aerosol infection as described in (E) and expressed as O.D. *n* = 6. Mean ± SEM is shown. (H) WT mice were immunized s.c. with MPLA + RBD-S1 and treated with α-IFN-γ blocking antibody (or isotype Ctrl) at day 0. dLNs were analyzed 7 days post infection. (I) Quantification of T_FH (left) and T_H1 (right), expressed as percentages out of endogenous effector CD4⁺ T cells in dLNs of s.c. infected mice described in (H). *n* = 8. Mean ± SEM is shown. Data are representative of three independent experiments. An unpaired two-tailed t test was applied. **p-value = 0.0054 (T_FH), ***p-value = 0.0001 (T_H1). (J) Quantification of Bcl-6⁺ B cells expressed as percentages out of endogenous B cells in dLNs of s.c. infected mice described in (H). *n* = 8. Mean ± SEM is shown. Data are representative of three independent experiments. An unpaired two-tailed t test was applied. ***p-value = 0.0003. Source data are available online for this figure.

# Methods

### Reagents and tools table

| Reagent/Resource | Reference or Source | Identifier or Catalog Number |
|---|---|---|
| **Experimental models** | | |
| C57BL/6N (*M. musculus*) | Charles River | C57BL/6NCrl |
| C57BL/6-Ly5.1 (CD45.1) (*M. musculus*) | Charles River | B6.SJL-PtprcᵃPepcᵇ/BoyCrl |
| CCR2⁻/⁻ (*M. musculus*) | The Jackson Laboratory | B6.129S4-Ccr2^tm1Ifc/J |
| IFNg-YFP (*M. musculus*) | The Jackson Laboratory | 129S4(B6)-Ifng^tm3-1Lky/J |
| IFNGR1⁻/⁻ (*M. musculus*) | The Jackson Laboratory | B6.129S7-Ifngr1^tm1Agt/J |
| IFN-g⁻/⁻ (*M. musculus*) | The Jackson Laboratory | B6.129S7-Ifng^tm1Ts/J |
| Mice bearing LCMV-specific transgenic CD4⁺ T cells (Smarta) (*M. musculus*) | Swiss Immunological Mouse Repository (SwImMR) | N/A |
| CD11c-DTR mice (*M. musculus*) | From M. De Palma and L. Naldini (San Raffaele Scientific Institute) | N/A |
| **Recombinant DNA** | | |
| **Antibodies** | | |
| Rat anti-mouse CD185 (CXCR5) (1:100) | BD Bioscience | 2G8 |
| Rat anti-mouse CD8a (1:100) | Biolegend | 53-6.7 |
| Rat anti-mouse T-bet (1:100) | Biolegend | 4B10 |
| Rat anti-mouse CD4 (1:100) | BD Biosciences | RM4-5 |
| Mouse anti-mouse/human Granzyme B (1:100) | Biolegend | GB11 |
| Mouse anti-mouse CD45.1 (1:100) | Biolegend | A20 |

| Reagent/Resource | Reference or Source | Identifier or Catalog Number |
|---|---|---|
| Rabbit anti-mouse TCF1/TCF7 (1:100) | Cell Signaling Tech | C63D9 |
| Rat anti-mouse/human CD11b (1:100) | Biolegend | M1/70 |
| Rat anti-mouse IFN-γ (1:100) | BD Pharmingen | XMG1.2 |
| Rat anti-mouse Ly-6G (1:100) | Biolegend | 1A8 |
| Mouse anti-mouse Bcl-6 (1:100) | BD Biosciences | K112-91 |
| Hamster anti-mouse CD11c (1:100) | BD Biosences | HL3 |
| Rat anti-mouse CD186 (CXCR6) (1:100) | Biolegend | SA051D1 |
| Mouse anti-mouse I-Aᵇ (1:100) | Biolegend | AF6-120.1 |
| Hamster anti-mouse CD183 (1:100) | BD Biosences | CXCR3-173 |
| Rat anti-mouse CD44 (1:100) | Biolegend | IM7 |
| Rat anti-mouse Ly-6C (1:100) | Biolegend | HK1.4 |
| Rat anti-mouse CD62L (1:100) | Biolegend | MEL-14 |
| Rat anti-mouse CD45R/B220 (1:100) | Biolegend | RA3-6B2 |
| Rat anti-mouse TCR V alpha 2 (1:100) | eBioscence | B20.1 |
| Mouse anti-mouse NK-1.1 (1:100) | Biolegend | PK136 |
| Hamster anti-mouse Vb 8.3 TCR (1:100) | Pharmingen | 1B3.3 |
| Rat anti-mouse CD335 (NKp46) (1:100) | Biolegend | 29A1.4 |
| Rat anti-mouse IFNGR1 (1:100) | BD Biosciences | GR20 |
| Hamster Anti-Mouse CD279 (PD-1) (1:100) | BD Biosciences | J43 |
| Anti-human/mouse/rat CD278 (ICOS) (1:100) | Biolegend | C398.4A |

| Reagent/Resource | Reference or Source | Identifier or Catalog Number |
|---|---|---|
| InVivoMab α-IFN-γ blocking antibody | BioXcell | Clone XMG1.2 #BE0055 |
| Rat IgG1 isotype control | BioXcell | Clone HRPN #BE0088 |
| InVivoMab α-IL-12 blocking antibody | BioXcell | Clone R2-9A5 #BE0233 |
| Rat IgG2b isotype control | BioXcell | Clone LFT-2 #BE0090 |
| InVivoMab α-NK1.1 depleting antibody | BioXcell | Clone PK136 #BE0036 |
| InVivoMab α-CD8 depleting antibody | BioXcell | Clone YTS 169.4 #BE0117 |
| Anti-CD16/32 antibody (1:100) | Invitrogen | #14-0161-82 |
| Rat anti-mouse Ter-119 | Biolegend | Ter-119 |
| LIVE/DEAD Fixable Aqua Dead Cell Stain | ThermoFisher Scientific | L34957 |
| LIVE/DEAD Fixable Near-IR Dead Cell Stain | ThermoFisher Scientific | L23105 |
| Goat α-human IgG Fc capturing Ab (1:1000) | Jackson Immunoresearch | #109005098 |
| HRP Goat anti-mouse IgG (1:500) | PerkinElmer | NEF822001EA |
| Anti-mouse IgM, IgG1 or IgG3 conjugated with horseradish peroxidase | SouthernBiotech | #5300-05B |
| **Oligonucleotides and other sequence-based reagents** | | |
| Ifng (Mm01168134_m1) | Thermo Fisher Scientific | 4351368 |
| Gapdh (Mm99999915_g1) | Thermo Fisher Scientific | 4351368 |
| **Chemicals, Enzymes and other reagents** | | |
| rVSV (a recombinant VSV expressing a GP derived from the LCMV WE strain and recognized by Smarta TCR-transgenic instead of the VSV GP) | Fallet et al, 2016 De Giovanni et al, 2020 | |
| rLCMV (a recombinant LCMV clone 13 expressing a GP derived from the LCMV WE strain and recognized by Smarta TCR-transgenic instead of the LCMV Cl13 GP) | Fallet et al, 2016 De Giovanni et al, 2020 | |
| Mouse-adapted SARS-CoV-2 strain (rSARS- N501YMA30) | From Stanley Perlman | |
| rRBD (Sars Cov-2 (2019-nCoV) Spike RBD Protein (S1 Subunit, FC Tag)) | Sino Biological | #40592-V02H |
| MPLA-SM* VacciGrade™ | InvivoGen | vac-mpla2 |
| Recombinant Mouse IL-7 | Bio-Techne | #407-ML-005 |
| Cas9/gRNA RNP complexes containing sgRNAs (either control or targeted to IFNGR1) and cas9 | Integrated DNA Technologies (IDT) | N/A |
| Antigen Fix | Diapath | #P0016 |
| KIllik – O.C.T. freezing medium | Bio-Optica | #05-9801 |

| Reagent/Resource | Reference or Source | Identifier or Catalog Number |
|---|---|---|
| FluorSave™ Reagent | Merck Millipore | #345789 |
| GP-1 from LCMV WE strain | N/A | N/A |
| Bovine Serum Albumin (BSA) Fraction V, US Origin, lyophilized powder | PAN-BIOtech | P06-1391500 |
| TMB Substrate Reagent set | BD Bioscience | #555214 |
| SARS-CoV-2 S1 subunit protein | RayBiotech | #230-30162 |
| Tween-20 | Thermo Scientific | #003005 |
| 0.3 M H$_2$SO$_4$ | N/A | N/A |
| Diphtheria toxin (DTX) | Millipore | #322326 |
| GP61–80 peptide from LCMV (GLKGPDIYKGVYQFKSVEFD) | N/A | N/A |
| Brefeldin A (GolgiPlug) | Sigma-Aldrich | B7651-25MG |
| Ammonium chloride (ACK) lysis buffer | N/A | N/A |
| **Software** | | |
| FlowJo Version 10.5.3 | Treestar | |
| Imaris | Bitplane | |
| GraphPad Prism software version 9.5 | GraphPad | |
| HISAT (version 0.1.6) | Kim et al, 2015 | |
| Seurat (v4.0.2) | Stuart et al, 2019 | |
| Cell Ranger (v.6.0.2) | 10x Genomics | |
| UMI-Tools (v.1.0.0) | Smith et al, 2017 | |
| STAR (v.2.5.3a) | Dobin et al, 2013 | |
| featureCounts (v.1.6.4) | Liao et al, 2014 | |
| Samtools software (v1.9) | Danecek et al, 2021 | |
| **Other** | | |
| CD4$^+$ T Cell Isolation Kit, mouse | Miltenyi Biotec | #130-104-454 |
| Primary Cell Optimization 4D-Nucleofector® X Kit | Lonza | N/A |
| CM1520 cryostat | Leica | |
| Inverted Leica microscope (SP8) | Leica Microsystems | |
| 96-well half-volume polystyrene plates | Corning | #3690 |
| Foxp3/Transcription Factor Staining Buffer set | eBioscience | #00-5523-00 |
| RPMI 1640 | SIAL S.R.L. | R8758-500ML |
| BD FACSCanto II | BD Bioscience | |
| BD FACSymphony A5 | BD Bioscience | |
| Cytek Aurora | Cytek Bioscience | |
| PBS 1X w/o Ca & Mg | SIAL S.R.L. | SIAL-PBS-2A |
| Triton™ X-100 | Sigma-Aldrich | T9284-100ML |
| Fetal Bovine Serum (FBS) | Corning | 35-079-CV |
| Illumina NextSeq 500 platform | Illumina | |

| Reagent/Resource | Reference or Source | Identifier or Catalog Number |
|---|---|---|
| MACSQuant Tyto Cell Sorter | Miltenyi Biotec | |
| Chromium platform (10x) using the Chromium Next GEM Single Cell 3′ v3.1 | Dual Index | |
| TapeStation instrument | Agilent | |
| NovaSeq 6000 platform | Illumina | |
| ReliaPrep RNA Miniprep system | Promega | |
| QuantStudio 5 Real-Time PCR system | Thermo Fisher Scientific | |
| HBSS | Corning | MDTC20-021-CVR |
| Inalation chamber | DSI Buxco Respiratory Solutions | |
| Superfrost™ Plus Microscope Slides | Thermo Scientific | #22037246 |

## Mice

All experimental animal procedures were approved by the Institutional Animal Committee of the San Raffaele Scientific Institute and by the Italian Ministry of Health (Authorizations #954/2020-PR and #971/2024-PR). Animals were handled in compliance with Institutional Committee and European ethical guidelines for animal care. Mice were housed under specific pathogen-free conditions and used at 8–10 weeks of age, unless otherwise indicated. In all experiments female mice matched for age were used. C57BL/6 and C57BL/6-Ly5.1 (CD45.1) (inbred C57BL/6) mice were purchased from Charles River. CCR2$^{-/-}$ (B6.129S4-Ccr2tm1Ifc/J), IFNg-YFP (129S4(B6)-Ifnγtm3-1Lky/J), IFNGR1$^{-/-}$ (B6.129S7-Ifngr1tm1Agt/J) and IFN-γ$^{-/-}$ (B6.129S7-Ifngtm1Ts/J) mice were purchased from The Jackson Laboratory. Mice bearing LCMV-specific transgenic CD4$^+$ T cells (Smarta) were obtained through the Swiss Immunological Mouse Repository (SwImMR). CD11c-DTR mice have been described previously (Jung et al, 2002) and were obtained from M. De Palma and L. Naldini (San Raffaele Scientific Institute). Bone marrow (BM) chimeras were generated by irradiation of C57BL/6 mice with ~900 rad and reconstitution with the indicated bone marrow; mice were supplied with antibiotic-supplemented water and allowed to reconstitute for at least 8 weeks prior to use.

## Infections and immunizations

Mice were infected s.c.ly (s.c.) in the footpad with $1 \times 10^5$ Plaque-Forming Unit (PFU) of rVSV (a recombinant VSV expressing a GP derived from the LCMV WE strain and recognized by Smarta TCR-transgenic instead of the VSV GP) or with $1 \times 10^5$ Focus-Forming Unit (FFU) of rLCMV (a recombinant LCMV clone 13 expressing a GP derived from the LCMV WE strain and recognized by Smarta TCR-transgenic instead of the LCMV Cl13 GP) (Fallet et al, 2016; De Giovanni et al, 2020). In indicated experiments, mice

were infected intravenously (i.v.) with $2 \times 10^5$ FFU of rLCMV. Viruses were propagated and quantified as described in previous studies (Kuka et al, 2012; Sammicheli et al, 2016; De Giovanni et al, 2020) and were diluted in 25 µl of HBSS before s.c. footpad injection. Viral titers from dLNs of LCMV-infected mice were measured by focus assay. Infection of C57BL/6 mice with aerosolized SARS-CoV-2 was performed as described (Fumagalli et al, 2021, 2024). Briefly, non-anesthetized mice were placed in a nose-only Allay restrainer on the inhalation chamber (DSI Buxco respiratory solutions; DSI). To reach a target accumulated inhaled aerosol (also known as delivered dose), C57BL/6 mice were exposed to a target accumulated inhaled aerosol of the mouse-adapted SARS-CoV-2 strain (rSARS-N501YMA30) kindly provided by Stanley Perlman. Primary inflows and pressure were controlled and set to 0.5 l min$^{-1}$ per port and −0.5 cmH2O, respectively. Infected mice were monitored daily to record body weight and clinical and respiratory parameters. All infectious work was performed in designated Biosafety Level 2 (BSL-2) and BSL-3 workspaces in accordance with institutional guidelines.

In immunization settings, mice were injected s.c. in the footpad with 5 µg of rRBD (Sars Cov-2 (2019-nCoV) Spike RBD Protein (S1 Subunit, FC Tag) Sino Biological #40592-V02H) conjugated to 10 µg of MPLA-SM* VacciGrade™ (vac-mpla2) in a volume of 30 µl/footpad.

## T cell isolation, adoptive transfer, and in vivo treatments

CD4$^+$ T cells were negatively selected from spleens of naive Smarta CD45.1$^+$ transgenic mice by magnetic isolation (Miltenyi Biotec), with purity always above 98% as determined by flow cytometry. Unless otherwise indicated, $0.5 \times 10^6$ Smarta T cells were injected i.v. into indicated recipients one day before infection. In indicated experiments, mice were treated with: InVivoMab α-IFN-γ blocking antibody (BioXcell Clone XMG1.2 #BE0055) or rat IgG1 isotype control (BioXcell Clone HRPN #BE0088): 250 µg intraperitoneally (i.p.) at day 0 (or in selected experiments at day 3); InVivoMab α-IL-12 blocking antibody (BioXcell Clone R2-9A5 #BE0233) or rat IgG2b isotype control, (BioXcell Clone LFT-2 #BE0090): 1 mg i.p. at day 0 and 3 after infection; InVivoMab α-NK1.1 depleting antibody (BioXcell Clone PK136 #BE0036): 1 mg i.p. at day 0 and day 1 after infection; InVivoMab α-CD8 depleting antibody (BioXcell Clone YTS 169.4 #BE0117): 200 µg i.p. at day −1 and day 2 after infection.

To deplete DCs 500 ng of diphtheria toxin (DTX, Millipore, #322326) diluted in 200 µl of PBS was administered i.p. one day before the infection and every other day thereafter to CD11c-DTR/IFN-γ$^{-/-}$ and CD11c-DTR/WT BM chimeras, respectively.

## CRISPR/Cas9-mediated IFNGR1 knockout in primary CD4$^+$ T cells

CRISPR/Cas9-Mediated IFNGR1 knockout was performed following the protocol in (Oh et al, 2019). Briefly, CD4$^+$ T cells were negatively selected from spleens of naive Smarta CD45.1$^+$ transgenic mice by magnetic isolation (Miltenyi Biotec), with purity always above 98% as determined by flow cytometry. Cells were then resuspended at a concentration of 10$^6$/ml and cultured with recombinant IL-7 (5 ng/ml) overnight at 37 °C. Cas9/gRNA

RNP complexes containing sgRNAs (either control or targeted to IFNGR1) and cas9 were assembled following the protocol provided by the supplier (Integrated DNA Technologies, IDT). CD4[+] T cells were resuspended in nucleofection buffer and transfected with Cas9/RNP complexes following the instructions of the Primary Cell 4D-Nucleofector X Kits (Lonza). Cells were then analyzed by flow cytometry to evaluate the transfection efficiency (through fluorescence conferred by ATTO550).

## Single cell suspensions and flow cytometry

Single-cell suspensions of spleens and LNs were obtained by mechanical dissection as previously described (Kuka et al, 2012; Sammicheli et al, 2016; De Giovanni et al, 2020; Fiore et al, 2023). Red blood cells were lysed with ammonium chloride (ACK) lysis buffer. In selected experiments, cell suspensions were plated in round-bottom 96-well plates ($1*10^6$ cells/well) and restimulated for 4 h with 2 μM GP61–80 peptide from LCMV (GLKGPDIYKGVYQFKSVEFD) in the presence of Brefeldin A (GolgiPlug, 1 ml/ml), in RPMI supplemented with 10% fetal bovine serum.

All flow cytometry stainings of surface-expressed markers were performed in FACS Buffer containing PBS and 2% FBS at 4 °C, while intracellular molecule staining was performed using Foxp3/Transcription Factor Staining Buffer set (eBioscience, #00-5523-00), following the manufacturer's instructions at room temperature. Anti-CD16/32 antibody (Invitrogen # 14-0161-82) was added to cell pellets prior to staining with fluorochrome-conjugated antibodies to block Fc receptors. Antibodies (Abs) used were purchased from BD Bioscience, Invitrogen, Biolegend or Cell Signalling and are indicated in the table below. Flow cytometry analyses were performed on BD FACSCanto II, BD FACSymphony A5 or Cytek Aurora and analyzed with FlowJo software (Treestar).

| Antibodies | |
|---|---|
| Rat anti-mouse CD185 (CXCR5) (2G8) | Rat anti-mouse CD8a (53-6.7) |
| Rat anti-mouse T-bet (4B10) | Rat anti-mouse CD4 (RM4-5) |
| Mouse anti-mouse/human Granzyme B (GB11) | Mouse anti-mouse CD45.1 (A20) |
| Rabbit anti-mouse TCF1/TCF7 (C63D9) | Rat anti-mouse/human CD11b (M1/70) |
| Rat anti-mouse IFN-γ (XMG1.2) | Rat anti-mouse Ly-6G (1A8) |
| Mouse anti-mouse Bcl-6 (K112-91) | Hamster anti-mouse CD11c (HL3) |
| Rat anti-mouse CD186 (CXCR6) (SA051D1) | Mouse anti-mouse I-A^b (AF6-120.1) |
| Hamster anti-mouse CD183 (CXCR3-173) | Rat anti-mouse CD44 (IM7) |
| Rat anti-mouse Ly-6C (HK1.4) | Rat anti-mouse CD62L (MEL-14) |
| Rat anti-mouse CD45R/B220 (RA3-6B2) | Rat anti-mouse TCR V alpha 2 (B20.1) |
| Mouse anti-mouse NK-1.1 (PK136) | Hamster anti-mouse Vβ 8.3 TCR (1B3.3) |
| Rat anti-mouse CD335 (NKp46) (29A1.4) | Rat anti-mouse IFNGR1 (GR20) |

## Confocal immunofluorescence staining

For confocal microscopy analysis LNs were directly collected and incubated for 90 min at room temperature in Antigen Fix (Diapath #P0016), and then washed in DPBS and dehydrated in 30% Sucrose at 4 °C. LNs were embedded in OCT freezing media (Killik Bio-Optica #05-9801) and 20 μm cryosections were prepared on a CM1520 cryostat (Leica), adhered to Superfrost Plus slides (Thermo Scientific) and stored at −20 °C. Sections were permeabilized and blocked with Blocking Buffer composed of DPBS, 10% FCS and 0.3% Triton X-100 (Sigma-Aldrich) and stained in the same buffer. Anti-CD16/32 antibody (Invitrogen # 14-0161-82) was added to cell pellets prior to staining with fluorochrome-conjugated antibodies to block Fc receptors. Before staining with fluorochrome-conjugated antibodies, slides were stained with Anti-mouse Fc Block antibody to block non-specific binding sites. The following fluorochrome-conjugated antibodies were used for cryosections staining: rat αB220 (RA3-6B2), mouse αCD45.1 (A20) and rabbit αTCF1/TCF7 (C63D9). Stained slides were mounted with FluorSave™ Reagent (Merck Millipore, #345789) and Images were acquired on an inverted Leica microscope (SP8, Leica Microsystems) with a motorized stage for tiled imaging using an HC PL APO CS 20X (NA 0.7) Dry or HCX PL APO λ blue 40X (NA 1.25) Oil objectives. To minimize fluorophore spectral spillover, we used the Leica sequential laser excitation and detection modality. B cell follicles were defined on the basis of B220 staining. For three-dimensional image acquisition, 6–10 xy stacks (1024 × 1024 pixels) sampled with 2-μm z spacing were acquired to provide image volumes that were 20 μm in depth.

## ELISA

The GP-1-IgG ELISA was carried out in 96-well half-volume polystyrene plates (Corning #3690). Plates were coated overnight at 4 °C with goat α-human IgG Fc capturing Ab (Jackson Immunoresearch #109005098) diluted 1:1000 in 0.1 M sodium carbonate buffer (pH 9.6). Afterward, the plates were blocked for 1 h with 5% milk diluted in PBS-Tween (0.05%). Thereafter, the plates were incubated with 50 μl per well of GP-1-IgG-containing cell supernatant for 1 h. Sera were diluted 1:4 in 5% milk in the first 96-well row and then 1:2 serial dilution were carried out in the GP-1-IgG-saturated plates, followed by incubation for 1 h. Finally, the plates were incubated for 1 h with HRP Goat anti-mouse IgG (PerkinElmer NEF822001EA) diluted 1:500 in 5% milk. HRP was detected by using TMB Substrate Reagent set (BD Bioscience #555214). All steps were carried out at room temperature. Between each step the plates were washed five times with PBS-T. Titers represent double-above-background values.

The SARS-CoV-2 S1 RBD-specific ELISA was carried out by coating plates with recombinant SARS-CoV-2 S1 subunit protein (RayBiotech, 230-30162) at a concentration of 2 μg/ml in PBS and incubated overnight (O/N) at 4 °C. Subsequently, the plates were blocked with PBS containing 1% bovine serum albumin (PBS-1% BSA) for 1 h at room temperature. The sera were then added at a dilution of 1/20 (sera from day 7) or 1/500 (sera from days 14, 21 and 28) and diluted 1:10 up to 1/1280 or 1/32,000, respectively, in duplicate, and the plates were incubated for 2 h at room temperature. After five washes with 0.05% Tween 20 in PBS, the

secondary anti-mouse IgM, IgG1 or IgG3 conjugated with horse-radish peroxidase (SouthernBiotech # 5300-05B) was added and the plates were incubated for 1 h at room temperature. After washing, the binding of the secondary antibody was detected by adding the substrate 3,3′,5,5′-tetramethylbenzidine (BD Biosciences). The reaction was blocked with 0.5 M $H_2SO_4$ and the absorbance at 450 nm and reference 630 nm was measured.

## Single-cell RNA sequencing (1)—library preparation

Single cell populations from rVSV-infected mice (CD4+CD45.1+ or CD4+ CD45.1+PD-1+ICOS+405 cells), rLCMV-infected mice or not infected SMARTA mice (CD4+CD45.1+) were sorted by using the following flow cytometry antibodies: APC-CXCR5 (2G8; BD Biosciences), APC-Cy7-CD45.1 (A20; Biolegend), eFluor450-CD4 (RM4-5; eBioscience), BV605-ICOS (C398.4A; Biolegend), PE-PD-1 (J43; eBioscience), AxFl488-B220 (RA3-6B2; Biolegend), AxFl488-NK1.1 (PK136; Biolegend), PE-Cy7-CD8a (53-6.7; Biolegend). LIVE/DEAD Fixable Aqua Dead Cell Stain (ThermoFisher Scientific) was used to exclude dead cells. Sorting was performed following exclusion of doublets, dead cells, and B220+B cells, NK1.1+NK cells, CD8a+T cells, and Ter119+erythrocytes.

Single-cell libraries were prepared as previously described (Jaitin et al, 2014). Briefly, mRNA from cells sorted into cell capture plates was barcoded, converted to cDNA and pooled with an automated pipeline. The pooled sample was then linearly amplified by T7 in vitro transcription, and the resulting RNA was fragmented and converted to a sequencing-ready library by tagging the samples with pool barcodes and Illumina sequences during ligation, reverse transcription and PCR. Each pool of cells was tested for library quality, and the concentration was assessed as described (Jaitin et al, 2014). RNA-seq libraries were sequenced on an Illumina NextSeq 500 platform, at a median sequencing depth of 15,054 reads per cell. Sequences were mapped to the mouse genome (mm10), demultiplexed, and filtered as previously described (Berglund et al, 2018), with the following adaptations. Mapping of reads was done using HISAT (version 0.1.6); reads with multiple mapping positions were excluded. Reads were associated with genes if they were mapped to an exon, using the UCSC genome browser for reference. We estimated a median of 2% spurious UMI in the data using statistics on empty MARS-seq wells.

## Single-cell RNA sequencing (2)—library preparation

CD4+ Smarta T cells from rLCMV-infected mice (day 5) treated or not with α-IFN-γ blocking antibody were sorted on MACSQuant Tyto Cell Sorter (Miltenyi Biotec) by using the following flow cytometry antibodies: PE-CD45.1 (A20; Biolegend), eFluor450-CD4 (RM4-5; eBioscience). LIVE/DEAD Fixable Near-IR Dead Cell Stain (ThermoFisher Scientific) was used to exclude dead cells. Sorting was performed following exclusion of doublets and dead cells.

Single cells were processed on the Chromium platform (10x) using the Chromium Next GEM Single Cell 3' v3.1 (Dual Index). After quality controls and quantification on TapeStation instrument (Agilent), libraries were sequenced on NovaSeq 6000 platform (Illumina) generating around 18,000 reads/cell. Raw sequencing

data were demultiplexed with the mkfastq application (Cell Ranger v.6.0.2). UMI-Tools (v.1.0.0) whitelist and extract commands were used to identify and select the number of cell barcodes to use in downstream analysis. Reads were mapped to the reference genome using STAR v.2.5.3a and assigned to genes with featureCounts v.1.6.4. GRCm38 was used as the reference genome. Bam files were sorted with samtools software (v1.9). Finally, Umi-Tools count was used to processing the UMIs aligned to each gene in each cell to find the number of distinct, error-corrected UMIs mapping to each gene. The UMI count tables of each cellular barcode were used for further analysis.

## Single-cell RNA sequencing bioinformatics analysis

Single cell data analysis was performed using Seurat (v4.0.2). Cells with sufficient bioinformatic quality were obtained after applying a filter of at least 200 genes expressed per cell and only genes expressed in at least 5 cells were retained. Moreover, cells with more than 10% of reads mapped to mitochondrial genes were also excluded from the analysis. UMI count matrix was further normalized and scaled following the standard Seurat workflow and Umap reduction was then applied on first 30 Principal Components after running PCA. Unbiased clustering was computed using the FindClusters function in Seurat with default parameters and a resolution value of 0.4. Specific markers for the different unbiased clusters were found using the function FindAllmarkers or FindMarkers in Seurat with default parameters. The plots showing normalized expression values with a color scale on top of Umap plots (on Figs. 1E and 2H,J; Appendix Fig. S2A) and the Violin plots of specific genes were produced with FeaturePlot and VlnPlot Seurat functions, respectively. The max.cutoff parameter is set to "q95". The gene signature in Fig. 2K was calculated with the AddModuleScore function in Seurat.

## qPCR

Total RNA was isolated from frozen LNs or spleens with the ReliaPrep RNA Miniprep system (Promega), following the manufacturer's instructions. One microgram of total RNA was reverse transcribed before qPCR analyses for *Ifng* (Mm01168134_m1) in a QuantStudio 5 Real-Time PCR System (all from Thermo Fisher Scientific). All experiments were done in duplicate, and data were normalized to the housekeeping gene *Gapdh* (Mm99999915_g1, Thermo Fisher Scientific).

## Statistical analyses

Flow and imaging data were collected using FlowJo Version 10.5.3 (Treestar) and Imaris (Bitplane), respectively. Statistical analyses were performed with GraphPad Prism software version 9.5 (GraphPad). Results are expressed as mean ± SEM. Means between two groups were compared with unpaired two-tailed t test. Means among three or more groups were compared with one-way or two-way ANOVA. Uncorrected Fisher LSD post-test was used for multiple comparisons. Significance is indicated as follows: *$p$-value < 0.05; **$p$-value < 0.01; ***$p$-value < 0.001; ****$p$-value < 0.0001. Comparisons are not statistically significant unless indicated.

## Data availability

The source data that support the findings of this study are openly available in the San Raffaele Open Research Data Repository at https://ordr.hsr.it/preview/c9r2fwjhr4?a=237335b5-8ab6-410e-bb9d-c29126a695bf. The scRNA-seq data shown in Fig. 1 and Appendix Fig. S2 are available in the Gene Expression Omnibus (GEO) database under accession no. GSE239968. The scRNA-seq data show in Fig. 2 and Appendix Fig. S7 are available in the Gene Expression Omnibus (GEO) database under accession no. Further information and requests for resources and reagents should be directed to and will be fulfilled by the lead contact, Mirela Kuka (kuka.mirela@hsr.it).

The source data of this paper are collected in the following database record: .

## Peer review information

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

## Acknowledgements

We thank M. Silva and M. Tinelli for secretarial assistance and the members of the Kuka and Iannacone Laboratory for helpful discussions. Flow cytometry was carried out at FRACTAL, a flow cytometry resource and advanced cytometry technical applications laboratory established by the San Raffaele Scientific Institute. Confocal immunofluorescence histology was carried out at Alembic, an advanced microscopy laboratory established by the San Raffaele Scientific Institute and the Vita-Salute San Raffaele University. We thank D. Pinschewer for rLCMV and Stanley Perlman for the mouse-adapted SARS-CoV-2 strain (rSARS-N501YMA30). ES, CL, and MN conducted this study as partial fulfillment of their PhD in the PhD program in Basic and Applied Immunology and Oncology at Vita-Salute San Raffaele University. This research was funded by the European Union - Next Generation EU, Mission 4 Component 1 CUP D53D23016440001. M.K. is further supported by the Italian Ministry of University and Research grants PRIN-2017ZXT5WR, SIR-RBSI14BAO5, PRIN-20209Y5YFZ and PE00000007 (INF-ACT), and by the Italian Ministry of Health (MoH) Grant GR-2021-12372615. MI is supported by European Research Council (ERC) Consolidator Grant 725038, ERC Proof of Concept Grant 957502, Italian Association for Cancer Research (AIRC) Grants 19891 and 22737, Italian Ministry of Health (MoH) Grant RF-2018-12365801, Italian Ministry for University and Research (Project no. PE00000007, INF-ACT), and sponsored research agreements from Gilead Sciences, Asher Biotherapeutics and VIR Biotechnology. CL is supported by Fondazione Prossimo Mio. LGG is supported by the Italian Ministry of University and Research grants PRIN-20224NMLXK, PRIN-P2022Z8HNC, Italian Ministry for University and Research Grants PE00000007 (INF-ACT), and a donation from FONDAZIONE SAME.

## Author contributions

**Eleonora Sala**: Formal analysis; Investigation; Visualization; Writing—original draft. **Maria Nelli**: Formal analysis; Investigation; Visualization; Writing—review and editing. **Chiara Laura**: Data curation; Formal analysis; Visualization; Writing—review and editing. **Pietro Di Lucia**: Investigation. **Cristian G Beccaria**: Investigation. **Elisa B Bono**: Investigation. **Marta Mangione**: Investigation. **Davide Marotta**: Formal analysis; Investigation. **Valentina Sperto**: Investigation. **Marta Grillo**: Investigation. **Leonardo Giustini**: Investigation. **Fabio Tosi**: Formal analysis. **Jia Nie**: Investigation. **Daehong Kim**: Investigation. **Giuliana Furiato**: Investigation. **Chiara Malpighi**: Investigation. **Eleonora Consolo**: Investigation. **Burkhard Becher**: Resources; Supervision. **Eyal David**: Data curation; Investigation. **Merav Cohen**: Investigation. **Amir Giladi**: Investigation. **Ido Amit**: Resources; Supervision. **Remy Bosselut**: Resources; Supervision. **Luca G Guidotti**: Resources; Supervision; Funding acquisition. **Matteo Iannacone**: Resources; Supervision; Funding acquisition; Writing—review and editing. **Mirela Kuka**: Conceptualization; Data curation; Formal analysis; Supervision; Funding acquisition; Visualization; Writing—original draft; Writing—review and editing.

Source data underlying figure panels in this paper may have individual authorship assigned. Where available, figure panel/source data authorship is listed in the following database record: biostudies:S-SCDT-10_1038-S44318-025-00414-3.

## Disclosure and competing interests statement

MI participated in advisory boards/consultantship for Asher Biotherapeutics, GentiBio, Clexio Biosciences, Sybilla Biotech, BlueJay Therapeutics, Bristol

Myers Squibb, Aligos Therapeutics and receives funding from Asher Biotherapeutics and VIR Biotechnology. LGG participated in boards, advisory boards and consultantships for Genenta Science, Epsilen Bio, Gilead Sciences, Antios Therapeutics, Aligos Therapeutics, Medicxi, Chroma Medicine and Ananda Immunotherapies. The other authors declare no competing interests.

