## [Peer Review File · The EMBO Journal]

T-cell-derived IFN- γ suppresses T follicular helper cell differentiation and antibody responses

Eleonora Sala, Maria Nelli, Chiara Laura, Pietro Di Lucia, Cristian Beccaria, Elisa Bono, Marta Mangione, Davide Marotta, Valentina Sperto, Marta Grillo, Leonardo Giustini, Fabio Tosi, Jia Nie, Daehong Kim, Giuliana Furiato, Chiara Malpighi, Eleonora Consolo, Burkhard Becher, Eyal David, Merav Cohen, Amir Giladi, Ido Amit, Remy Bosselut, Luca Guidotti, Matteo Iannacone, and Mirela Kuka

Corresponding author(s): Mirela Kuka (kuka.mirela@hsr.it) , Matteo Iannacone (iannacone.matteo@hsr.it)

Review Timeline:

Submission Date:	17th Sep 24
Editorial Decision:	2nd Oct 24
Revision Received:	24th Jan 25
Editorial Decision:	19th Feb 25
Revision Received:	23rd Feb 25
Accepted:	3rd Mar 25

Editor: Ioannis Papaioannou

Transaction Report:

The manuscript was originally peer-reviewed in another journal.

Reviewers' Comments:

Summary of the findings

Reviewer #1: Sala et al., explored heterogeneity in the differentiation process of Th1 and Tfh CD4+ cells following subcutaneous LCMV infection. They demonstrate that enhanced Th1 differentiation of transferred CD4+ T cells, while Tfh formation was impaired. Using scRNAseq, the authors demonstrate a transitional Th1/Tfh precursor population that converts to Th1 cells via IFN γ . By blocking IFN γ early in LCMV infection, they showed that IFN γ plays a critical role in TH1 differentiation, and that Tfh differentiation is enhanced in the absence of IFN γ . In additional experiments, they identified T cells as the source of IFN γ driving Th1 differentiation.

Reviewer #2: In the manuscript "IFN- γ Suppresses T Follicular Helper Cell Differentiation and Antibody Responses", Sala et al. investigated the mechanism by which subcutaneous LCMV infection promotes a Th1 response and inhibits Tfh. They find that blockade of IFN γ is sufficient to increase Tfh polarization early after infection, with increases in Bcl6+ B cells, and increases in GP-binding antibody. This increase was associated with greater proportions of TCF-1+ SMARTA. IFN γ blockade at the time of s.c. infection (versus day 3) had greatest impact. The authors then to show that IFN γ blockade also increases Tfh differentiation in a model of s.c. immunization, whereas blockade during IV LCMV infection or s.c. rVSV infection did not have an impact on Tfh frequencies at day 5 or 7.

Reviewer #3: This review has been structured in accordance with the universal principled template outlined by Krummel et al (doi.org/10.1016/j.cell.2019.11.029).

In this manuscript by Sala et al, the authors seek to understand how LCMV delivered subcutaneously alters CD4 T cell differentiation. Specifically, the authors report that subcu LCMV induces high levels of IFN γ (higher than IV infection) leading to Th1-predominant differentiation, but with a subset of Tcf1hi cells whose expression of CXCR5 increases following IFN γ blockade, coupled with greater abundance of the Tcf1hi cells in the LN including in the B follicle with concomitant increase in GC-B cells at 7 days. The Tcf1 population did not rebound as strongly if the IFN γ blockade was induced late (at 3 days) instead of at infection. Overall, these studies are technically well-done and provide a mechanistic basis for understanding effects of early IFN γ , differences between routes of infection with LCMV, and the role of IFN γ in CD4 T cell differentiation.

We thank the three reviewers for the critical assessment of our work. To aid the readers in the understanding of the proposed biological model, we added a graphical abstract as Supplementary Figure 17.

Reviewer #3 (continues)

Experiments: Experiments are shown with the proper controls for each experiment. Flow cytometry plots show attention has been paid to antibody optimization and signal-to-noise discrimination.

Major: none

Minor: none

Completeness: Data shown support the contentions made in the title and abstract. The sequence of experiments is laid out well and the findings are generally consistent throughout. The confocal microscopy studies showing increased presence of Tcf1+CD45.1 cells in and around the B follicle with aIFN γ which complements the flow cytometry findings.

We thank Reviewer #3 for the overall positive assessment of our work.

Major:

1. The evidence demonstrating that the Tcf1hi cells have Tfh-relevant characteristics is based on

protein expression of CXCR5 and early indication of increased germinal center activity through GC-B cell frequencies. It would be helpful to see additional evidence of Tfh commitment. For example, further interrogation of Tfh signatures in the scRNAseq data might help determine commitment to the Tfh pathway at d5.

We added a GSEA analysis panel (new Fig. S2D) showing that the transcriptional signature of cluster 1 (Tcf-1⁺ cells) is enriched in genes expressed by CD4⁺ T cells polarized into T_{FH} upon rVSV infection. We believe this answers to the reviewer's comment suggesting that Tcf-1⁺ cells have an overall signature that resembles T_{FH} cells.

Later time points to more robustly show germinal centers could be considered.

We have added new data on Smarta CD4⁺ T cell polarization 14 days after infection in Figure 5, panels I-K. As discussed in the text, "The frequency of Smarta cells at this time-point was highly variable and significantly lower than day five, due to the contraction phase all T cells undergo after viral clearance (Fig. 5J). Nonetheless, IFN- γ blockade still resulted in increased T_{FH} frequencies (Fig. 5K) indicating that a single-dose of blocking antibody at the beginning of infection was sufficient to affect T cell polarization for weeks". Unfortunately, a substantial effect at later time-points was not observed in GC B cells, probably due to the fact that LCMV adopts many different mechanisms to interfere with GC responses (Kuka and Iannacone, 2018). In another setting, two weeks after Sars-CoV-2 infection, we observed a trend in the increase of IgG1 and IgG3 antibodies, which have been reported to be the only TFH-dependent isotypes (Chen et al., 2022) (Fig. 8G).

Or co-transfer of the CD4 treated with IFN γ at day 0 and day 3 which might reveal GC engagement by the day 0 but not day 3-treated cells.

If we understand the reviewer's suggestion correctly, they propose pre-treating naïve CD4⁺ T cells or CD4⁺ T cells sorted after three days of LCMV infection with IFN- γ , and transfer them back to new recipients. However, this assumes that CD4⁺ T cells respond to IFN- γ , which we found is not the case. Indeed, by using genetic models and Crispr/Cas9 technology, we found that the IFN- γ responsible for T_{FH} suppression is T cell-derived but does not act on CD4⁺ T cells themselves. Instead, IFN- γ is sensed by DCs, which might then acquire a phenotype that interferes with T_{FH} differentiation.

Protein expression for Bcl6 in the T cells at 7d would be a nice adjunct to the B cell data in Fig. 3D, if available.

We added a representative plot and a quantification of the frequency of endogenous Bcl-6⁺ CD4⁺ T cells at day 7 upon infection (new Fig. 4C and D).

Minor:

1. Sex of the mice used should be provided.

We specified in the methods section that we used female mice. Page 24 highlighted in yellow.

2. Flow cytometry plots to support Fig. 4 would be helpful (i.e. for 4C, 4F, and 4G).

Old Figure 4 (the one the reviewer is referring to) is now new Figure 5. As requested by the reviewer we added representative flow cytometry plots related to Figures 5C, 5F and 5G as supplementary Figure S7 (Figure S7A, B and C).

3. Gating scheme used to identify immune subsets in Fig 5A should be shown.

Old Figure 5 (the one the reviewer is referring to) is now new Figure 6. As requested by the reviewer we added the gating strategy related to Figure 6A as supplementary Figure S8.

Reproducibility: The statistical analysis section of the methods indicates experiments have been

performed in independent replicates and this is indicated per experiment. An appropriate number of mice were used. Statistical analyses involving >2 comparisons were performed as parametric ANOVA with post-test and two-sample analyses were performed by unpaired t-test. Clones of Ab are indicated for flow and immunofluorescence plots.

We thank Reviewer #3 for the overall positive assessment of our work.

Major:

1. Are the colors in S5A correct? Fig. 2I indicates cluster 0 (top right cluster) is 97% comprised of Ctrl but S5A appears as though nearly all of those cells are blue (indicated as aIFNg).

We thank the reviewer for catching this mistake, which was corrected in the new version of the figure (now new Figure S6A).

Minor:

1. Please consider moving Tables to supplemental and sorting tables 1 and 2 by Padj (as was done for Table 3) to improve readability.

We think it won't be helpful to sort tables 1 and 2 as suggested because upregulated and downregulated genes will be mixed. Thus, we would like to leave tables 1 and 2 like they are.

2. In Figure 1K, use of overlaid filled histograms makes it difficult to assess the individual samples. Use of offset or unfilled histograms might help the individual replicates show through.

We changed the histograms in Figure 1K as suggested by the reviewer.

3. The text indicates modest differences for Bcl6, Cxcr5, and Cxcr3 in Fig. S3b but only Cxcr5 is shown. It would help to show the direct comparison of Tcf1+ vs GzmB+ cluster for these proteins (or genes).

We think that the reviewer might have misunderstood figure S3. In panel A we show differences between Tcf-1+ and GzmB+ clusters for all markers mentioned in the text (therefore including Bcl6 and Cxcr3). Figure S3B was showing CXCR5 levels on total B cells compared to Tcf-1+ cells. To avoid any misunderstanding, we deleted panel B (and any related comment in the revised text) and discussed only panel A, now renamed Figure S3.

4. For Fig. S5B, statistical comparisons would be helpful.

We added statistical comparisons to the figure (new Fig. S6) as requested.

5. Time point for measurement of Ab in Fig 3F is unclear. The legend refers to "(H)" but it is unclear what panel this is referring to.

We specified the timepoint of Ab detection (day 7 upon infection) in the new version of the figure legend (new Figure 4G).

6. The double-break in Fig 5A at 72h is not clearly needed and makes it more difficult to compare the 72h frequencies to the 24h and 48h. The y-axis scale should be harmonized across all three panels.

Old Figure 5 (the one the reviewer is referring to) is now new Figure 6. As requested by the reviewer we changed the y-axis scale to be the same across the three panels.

Scholarship Appropriate literature is cited in the Introduction and Discussion to support the work.

Major:

1. None identified

Minor:

1. Other studies have evaluated the effects of IFN γ on B cells. For example, one possibility is that blockade of IFN γ limits differentiation of B cells including to ASC (i.e. PMID 31090539, 31076359, 31900342, 28554560, 7523492). Inclusion of these references in the introduction/discussion might help put the B cell findings in context.

We cited and discussed all the suggested references in the revised paper (please see page 19 in the Discussion session)

Significance

How do these findings advance the thinking in the field? If there are concerns about conceptual advance (e.g., if the advance is limited by previous work), please provide primary references.

Reviewer #1: As described by the authors, these results are in line with previously published work (both cited and un-cited). Further the authors overstate the significance and application to vaccinology, as they demonstrate a limited effect of IFN γ blockade on T_{fh} differentiation in models where they already dominate over Th1 cells.

We respectfully disagree with Reviewer 1's concern regarding the level of advancement and novelty of this research. We believe this study significantly enhances our understanding of the delicate balance between CD4⁺ T cell subsets during viral infections, with IFN- γ serving as a molecular switch that promotes cellular responses while suppressing T_{fh} responses. Notably, the IFN- γ involved is produced by the T cells themselves, suggesting a potential cross-antagonism between T_{H1} and T_{fh} cells—similar to the relationship between T_{H1} and T_{H2} cells—a concept that, to our knowledge, has not been previously reported. In addition, by using genetic models and Crispr/Cas9 technology, we found that the IFN- γ responsible for T_{fh} suppression is T cell-derived but does not act on CD4⁺ T cells themselves. Instead, IFN- γ might be sensed by DCs, which might then acquire a phenotype that interferes with T_{fh} differentiation. This is quite novel, since previous studies reporting a role for IFN- γ in T_{H1} maintenance have suggested an autocrine mechanism (Bradley et al., 1996; Lighvani et al., 2001; Whitmire et al., 2005). In the revised manuscript, we will clarify these ideas further for the benefit of both the reviewers and the readers.

Reviewer #2: This work highlights the extent to which early events after infection can dramatically impact CD4 T cell differentiation. There are several limitations to the work that limit the scope of impact and interpretation of results.

We are aware that limitations of this work exist, however we believe this study significantly enhances our understanding of the delicate balance between CD4⁺ T cell subsets during viral infections. Please see our point-by-point response to the reviewers' major concerns.

Reviewer #3: (See above)

We have addressed all major and minor concerns above given the format of the review.

Major concerns and limitations

List all concerns with the experimental and/or analytical approach(es) and data in the study, including the statistical analyses. Flag the 3 points (with an asterisk or another symbol) that you consider of central importance. If conceptual advance is a main concern, this can be indicated here as well as in above in the Significance questions.

Reviewer #1:

1. The key findings of this study are consistent with multiple publications and offer little additional insight. Thus, there is limited novelty of this work, given the literature has previously established:

- That the concept of balance between Th1 and Tfh differentiation is well established. The author cite some of these foundational papers and also reviews that establish this concept.

We have now added in the introduction “previous work has shown some degree of overlap or competition between these two CD4+ T cell subsets” (Nakayamada et al., 2011; Lönnberg et al., 2017). For example it was reported that T_{FH} and T_{H1} share a transitional phase expressing both T-bet and Bcl-6. While the cells progress into reinforcing T_{H1} phenotype, T-bet suppresses further T_{FH} differentiation by competing with Bcl-6. This competition seems to be cell-intrinsic since CD4⁺ T cells lacking T-bet differentiate into T_{FH} (Nakayamada et al., 2011; Lönnberg et al., 2017). In another study, Bcl6-expressing T_{FH} cells generated upon viral infection expressed T-bet, which was critical for their development and function and transcriptionally required for proper T_{FH} cell programming (Weinstein et al., 2018)”

We have also discussed this in the discussion “Thus, we propose that T_{H1} antagonize T_{FH} not only through competition of cell-intrinsic transcription factors as proposed by others (Nakayamada et al., 2011), but also through a cell-extrinsic effect on the surrounding microenvironment”

- A common precursor for Th1/Tfh that is regulated by IFN γ has previously been identified. Ref 21 (Nakayamada Immunity 2011).

We have cited Nakayamada’s paper. This paper demonstrates that a common precursor expressing T-bet can give rise to both T_{H1} and T_{FH} cells. That study describes an intrinsic competition between T-bet and Bcl-6 within the cell. We report that the competition between these two subsets appears to be cell extrinsic, mediated by IFN- γ produced by CD4⁺ T cells and acting on a cell type different from CD4+ themselves

- The reliance of IFN γ for full Th1 differentiation and inhibition of Tfh differentiation is known. Refs 12, 13, 46 (Zhang JEM 2001; Bradley JI 1996; Ryg-Cornejo Cell Rep 2016).

Zhang’s and Bradley’s papers highlight the role of IFN- γ in inducing T_{H1} cells but do not address its relationship with T_{FH} cells. While we acknowledge the relevance of Ryg-Cornejo’s data to our project, their study was conducted in the context of a parasitic infection (malaria), and the IFN- γ (along with TNF- α) required for T_{FH} suppression was not shown to be produced by T_{H1} cells themselves. We discuss this at page 19 as follows “The inhibitory effect of IFN- γ on T_{FH} differentiation and humoral immunity is not unprecedented. For instance, in severe malaria infection, a setting where T_{H1}-like dysfunctional T_{FH} cells expressing T-bet are observed, concomitant blockade of IFN- γ and TNF- α resulted in enhanced T_{FH} differentiation and improved antibody responses (Ryg-Cornejo et al., 2016). Nevertheless, in this context, the cells responsible for producing or sensing IFN- γ were not identified, and the underlying mechanism remained unexplored”

- CD4+Tcf1+ Precursor of Th1/Tfh – Xia Immunity 2022 (PMID: 35637103)

We have cited and discussed this paper on page 20 of the revised manuscript. Specifically, we note: ‘Recently, a population of Tcf-1+ Bcl-6- PD-1+ CD4⁺ T cells capable of differentiating into both effector and T_{FH} cells was described in the context of chronic LCMV infection and persistent antigen exposure (Xia et al., 2022). While we cannot entirely rule out that this population may functionally resemble the Tcf-1+ T-bet+ CD4⁺ T cells identified in our study, it is important to note that subcutaneous LCMV infection is typically cleared within 7-8 days, representing an acute infection model (Sammicheli et al., 2016).

- Precursor of Th1/Tfh independent of IL-12 – Krueger Immunity 2021 (PMID: 33773107)

Our understanding of the cited paper is that IL-12 is required for TH1 differentiation only in a later stage, and only for certain pathogens (e.g., Salmonella but not Influenza). This aligns with findings from our study and others {Oxenius et al., 1999, #27773; Schijns et al., 1998,

#9534}, which show that some viral infections can induce T_H1 responses independent of IL-12. While we are not the first to report this, we believe it is important to highlight this point, as it suggests the existence of other, yet unidentified, T_H1 -polarizing factors in viral infections.

- T cell derived IFN γ impacting B cell responses. Ref 51, 52 and PMID: 37699392 (Rubtsova JCI 2017; Obeng-Adjei PlosPath 2017; Arroyo-Díaz Immunity 2023).

As mentioned in the discussion, we are aware of the reported role of IFN- γ in inducing T-bet expression in B cells. We have discussed this at page 21 as follows “A role for IFN- γ on B cell activation has been reported by others although with different outcomes depending on the context (Abed et al., 1994; Myles et al., 2017; Obeng-Adjei et al., 2017; Unger et al., 2018; Stone et al., 2019; Zumaquero et al., 2019; Chodiseti et al., 2020; Arroyo-Díaz et al., 2023). In particular previous literature has suggested that IFN- γ can act directly on B cells to induce T-bet expression (Obeng-Adjei et al., 2017; Stone et al., 2019; Zumaquero et al., 2019; Chodiseti et al., 2020). T-bet-expressing B cells, observed in various chronic infections, severe malaria, and autoimmune diseases, are often characterized as 'atypical' due to their dysfunctional traits and pathogenicity (Obeng-Adjei et al., 2017; Rubtsov et al., 2017; Rubtsova et al., 2017; Burton et al., 2018). However, in the context of respiratory viral infections, T-bet-expressing T and B lymphocytes function collaboratively, facilitating optimal antiviral immunity (Mendoza et al., 2021). In our experimental setting IFNGR1 expression on B cells was not very high in the first three days upon infection, therefore we believe it is likely that the increase in Bcl-6⁺ B cells may result from enhanced T_{FH} differentiation following early IFN- γ blockade, whereas the decrease in T-bet⁺ B cells could be due to IFN- γ action directly on B cells at later time-points”

2. The authors claim universality of their findings, however, the persuasiveness of the supportive data is questionable. This is underlined by the inability to replicate findings in the context of LCMV i.v. infection.

The reviewer is correct in noting that IFN- γ 's role in suppressing T_{FH} cells has minimal effects in systemic infections. As explained in the paper, we believe this is due to the low levels of IFN- γ produced in such settings. We see this as a strength of our study, as it emphasizes that only when CD4⁺ T cells produce high levels of IFN- γ —such as in strong T_H1 polarization—does suppression of T_{FH} cells occur. We will work to clarify this point further in the revised manuscript.

Therefore, author's proposed application to vaccinology is overstated, as there is minimal impact of IFN γ blockade in settings where Tfh dominate over Th1, and no effect in a vaccine setting, unless IL-6 blockade is added.

The reviewer is correct in noting that IFN- γ has minimal impact in settings where T_{FH} cells dominate over T_H1 cells, likely due to low levels of IFN- γ in those contexts. However, the response can shift towards either T_H1 or T_{FH} polarization depending on the adjuvant used in vaccination. It is crucial to understand that when IFN- γ is produced at high levels, it can suppress humoral responses to infections or vaccines.

Regarding IL-6 blockade, our preliminary data suggest a potential antagonism between IL-6 and IFN- γ . However, based on the reviewer's feedback, we realize that introducing this added complexity may detract from the clarity of our message. Therefore, we have replaced the VSV model in the current version of the manuscript with a more relevant SARS-CoV-2 infection model, for which we have already conducted experiments.

Further, the authors state in the discussion that these results are most relevant when IFN γ is overexpressed, particularly in T cell transfer models (line 335-36), a situation that does not occur for humans.

We kindly disagree with the reviewer's conclusion, since we have shown that upon s.c. infection IFN- γ plays a role in suppressing T_{FH} also in the endogenous setting, a situation which can occur in humans.

Reviewer #2:

1. The mechanism by which IFN- γ inhibits Tfh development is not elucidated. Does IFN- γ act directly on CD4⁺ T cells and inhibit the Tfh lineage commitment? Or alternatively, does IFN- γ act on other cell types that facilitate Tfh differentiation? Is this a generalizable mechanism? The authors may be integrating this with the mechanisms published in ref 19, but these data and/or discussions are missing from the manuscript as is.

As suggested by the reviewer we have performed experiments aiming at elucidating the mechanism by which IFN- γ inhibits T_{FH} development. Notably, by using genetic models and Crispr/Cas9 technology, we found that the IFN- γ responsible for T_{FH} suppression is T cell-derived but does not act on CD4⁺ T cells themselves. Instead, IFN- γ is sensed by DCs, which might then acquire a phenotype that interferes with T_{FH} differentiation. This is quite novel, since previous studies reporting a role for IFN- γ in T_{H1} maintenance have suggested an autocrine mechanism (Bradley et al., 1996; Lighvani et al., 2001; Whitmire et al., 2005). The new data are reported in New Figures 7 and S14.

2. The authors do not show the impact of IFN γ on viral load after sc infection. As a result, it is difficult to discern whether the IFN γ blockade impacts the CD4 T cells directly or whether it impacts viral load and the viral load instead impacts CD4 T cell differentiation. The vaccine study in Figure 6J normalizes antigen load, but the effect size in this experiment is small and no antibody response is shown.

As suggested by the reviewer, we have measured viral loads in the LNs of mice 3 and 5 days after s.c. infection with LCMV. Notably, we found that the effect of IFN- γ blockade in CD4⁺ T cell polarization was not a result of higher viral replication, since viral titers in mice receiving the IFN- γ neutralizing Ab were slightly lower at day 3 and similar to control mice at day 5 after LCMV infection (New Fig. S5B). In addition, we have previously shown that CD4⁺ T cells differentiate into T_{H1} but not T_{FH} even when animals are infected with 100-fold higher or lower viral loads (De Giovanni et al., 2020).

3. The authors focus on very early, pre-GC events after infection. It is therefore unclear whether the impact on CD4 T cell differentiation is sustained or whether Tfh and GC development is instead delayed in control groups. If early events are the only focus of the manuscript, the conclusions should be narrowed to avoid miscommunication. Likewise, the impact on antibody development is a major feature of the manuscript's conclusions. However, this reviewer could only find GP-binding as an antibody readout. Is there a difference in neutralizing antibody early and late after infection?

We have added new data on Smarta CD4⁺ T cell polarization 14 days after infection in Figure 5, panels I-K. As discussed in the text, "The frequency of Smarta cells at this time-point was highly variable and significantly lower than day five, due to the contraction phase all T cells undergo after viral clearance (Fig. 5J). Nonetheless, IFN- γ blockade still resulted in increased T_{FH} frequencies (Fig. 5K) indicating that a single-dose of blocking antibody at the beginning of infection was sufficient to affect T cell polarization for weeks." As for antibodies, the ELISA we have performed highlights only antibodies that bind to LCMV GP and thus is highly specific. Unfortunately the titers are too low for the neutralization test to be performed.

4. The authors posit that IFN γ blockade only works in settings of high IFN γ , but they do not show the IFN γ levels in the other models displayed in Figure 6. This limits the ability to interpret results.

We measured IFN γ levels also in the immunization setting and data are shown in Fig. S16I.

5. The authors state that Figure 6I demonstrated 'marginal' increases in Tfh during IFN γ blockade at the time of rVSV infection, but the data do not support any increase and statistics for this conclusion are not shown. Why would there be no difference in this sc infection model?

We believe the VSV infection model behaves differently from LCMV due to the early induction of IL-6 in VSV infections. Our preliminary data suggest a potential antagonistic effect between IL-6 and IFN- γ . However, based on the reviewer's feedback, we realize this added complexity may detract from the clarity of our message. Therefore, we have replaced the VSV model in the manuscript with a more relevant aerosolized SARS-CoV-2 infection model, which better reflects human infection scenarios (Fig. 8E-G).

Reviewer #3: (See above)

Minor points and recommendations

Please list any additional comments and/or suggestions.

Reviewer #1:

1. IFN γ can be modified post-transcriptionally (Lee Immunity 2012, PMID: 23159227) and therefore, should be quantified at the protein level in Figure 6, and also for confirmation of the transcriptional IFN γ -YFP reporter in Figure 5.

We did our best to address this point; unfortunately there were technical challenges with the experiment, due to the low amount of material derived from popliteal LNs and we don't have conclusive results.

2. Line 278 typo – should be IL-6 dependent

We thank the reviewer for catching this typo, but since we took the VSV experiments out from the paper this will not be a problem anymore.

3. Line 317 states that it is unclear if IFN γ acts directly on B cells, however this should reference PMID: 37699392, which demonstrates IFN γ acts directly on B cells.

We have cited this and other references reporting the role of IFN- γ on B cells. Our statement "that it is unclear if IFN- γ acts directly on B cells" is referred to our own data as discussed at page 21 "In our experimental setting IFNGR1 expression on B cells was not very high in the first three days upon infection, therefore we believe it is likely that the increase in Bcl-6⁺ B cells may result from enhanced T_{FH} differentiation following early IFN- γ blockade, whereas the decrease in T-bet⁺ B cells could be due to IFN- γ action directly on B cells at later time-points"

4. The duration of antibody blockade (i.e. how long treatments last) should be evaluated for the interpretation of the timed inhibition studies in Figs 2, 4, and 6.

The blocking antibody was given at a single dose on day 0, and the effects on CD4⁺ T cell polarization were observed up to day 14 (Figure 5). However, when we gave the treatment at day 3, we did not observe T_{FH} rescue anymore (Figure 5), suggesting that the long-term effect on CD4⁺ T cell polarization is more likely to be attributed to the early CD4 T cell differentiation than to the duration of IFN- γ -blockade.

Reviewer #2:

1. The first half of the abstract makes several statements about antibody versus cellular immunity that are possibly too broad and conclusive.

We have improved the abstract to make sure it reflects the data presented.

2. Experiment for Figures 1F and H – There is no control here for CXCR5 staining. This is relevant as all other figures using this or nearly identical experimental design (Figures 2E/F, 4D, 5B/D show either more CXCR5+ staining in the control either by flow plot or by frequency of Tbet-CXCR5+ cells. A bar/dot summary plot for the data in Figure 1F would help reinforce the conclusion drawn in the first experiment – especially given that the frequency of Tbet-CXCR5+ cells in the control varies in the above-mentioned figures/experiments.

We added the requested CXCR5 control staining in new Figure S3C and D. In addition, we added as Figure S3B a quantification of T-bet⁺ and CXCR5⁺ cells for data in Figure 1F.

3. This work clearly builds on the data from reference 19. However, the manuscript does not stand alone in its discussion of the relevance of this question and where the gaps in the field are. In particular, the prior literature on the impact of IFN γ and Tfh/antibody/GC response is underdeveloped in the introduction. It is therefore difficult for the reader to quickly understand what is new and important in the presented findings.

We thank the reviewer for the suggestion, substantially changed the introduction, and mentioned the study on IFN- γ and T_{FH} in lines 80-85.

4. Does this rLCMV with a sc route behave like a chronic viral infection? How might that impact the findings?
5. *s.c. rLCMV infection is completely cleared from the LNs within 7-8 days of infection and does not establish chronicity (as shown in Sammicheli et al. Science Immunol 2016). We mention this also at page 20-21 of the Discussion “We cannot formally exclude that this population functionally resembles Tcf-1⁺T-bet⁺ CD4⁺ T identified in our setting: however, it is worth mentioning that the s.c. LCMV infection is cleared within one week and therefore represents an acute infection setting (Sammicheli et al., 2016)”*

6. The gating strategies for parent populations of the T-bet vs. CXCR5+ flow plots are not shown – it would improve clarity to show them.

We added the requested gating strategies in new Figures S1 and S3A-E.

7. The tables in the supplementary materials are not well labeled. It is hard to tell the two populations upon which the differential gene expression analysis is conducted from every table.

We have revised the tables clarifying the different conditions that are shown and using color code when possible.

Reviewer #3: (See above)

Clarity of reporting

Please comment on whether the paper adequately reports methods, number of samples and independent experiments, statistics, data, and if applicable, code.

Reviewer #1: Some questionable reporting over the total "n" was in experimental repeats with some experiments showing 3 points per group and others showing up to 9.

The variability of n in the mentioned experiment (new Fig. 6A) is due to the fact that several experiments were pooled because of the technical limitations in detecting IFN γ -YFP, and not all immune cell subsets could be stained at the same time.

Throughout % of cell populations are given, without any reporting of total cell numbers.

We have provided total cell numbers for Figure 2 (the first figure showing the effect of IFN- γ in CD4 T cell polarization). New data are reported in New Fig. S6. In general, throughout all the paper percentages reflect absolute numbers.

Numbers of "n" for imaging in Figure 5 is not indicated. Unclear what is was quantified from these images - an entire lymph node slice, multiple slices, or regions of interest shown. Unclear if there was a blinded approach taken to determine what regions to quantify.

We believe the reviewer refers to Figure 3 (the only figure that contains confocal images). The n of the slices was indicated already in the original version of the paper and is n=4

Reviewer #2: Adequate

Reviewer #3: (See above)

Dear Mirela, dear Matteo,

Thank you again for transferring to The EMBO Journal for our consideration your manuscript (EMBOJ-2024-119046) along with your detailed response to the comments of the reviewers who previously assessed it at another journal, as well as your provisional revision plan. As we have previously discussed, I have shared your manuscript and your point-by-point response with an additional arbitrator with familiarity both with the field and our journal and its scope, and I have now received their advice (included below).

As you will see, our arbitrator agrees with the original referees that the study is technically of high quality, and they further recognize that it is interesting and provides useful insights, but also points out that the overall conceptual advance of the manuscript as it stands remains modest, and finds some of your arguments in response to the reviewers not fully supported by the presented data. Our arbitrator further advises that the suggested revisions should be suitable for strengthening the manuscript significantly and highlights particular points which, in their view, are necessary to be addressed for the manuscript to be suitable for publication in The EMBO Journal.

In light of the original referee reports, your responses to them, and the additional input from our arbitrator, which we find fair and balanced, I would like to invite you to submit a revised version of your manuscript along the lines you describe in your revision plan and with particular attention to the points highlighted by our arbitrator. Please also submit a detailed point-by-point response to the previous referees' comments as well as our arbitrator's advice, describing all additions and changes to the manuscript. I should add that it is The EMBO Journal policy to allow a single round of major revision, and acceptance of your manuscript will therefore depend on the completeness of your responses in this revised version. Please let me know if you have any questions or comments that you would like to discuss with me.

We generally allow three months as standard revision time (January 1st, 2025), but we may be able to grant an extension to allow enough time for the revision should the need arise. Should you foresee a problem in meeting the three-month deadline, please let us know. As a matter of policy, competing manuscripts published during this period will not negatively impact our assessment of the conceptual advance presented by your study. However, we request that you contact us as soon as possible upon publication of any related work, to discuss how to proceed.

Thank you again for the opportunity to consider your work for publication in The EMBO Journal. I am looking forward to your revision.

Best regards,

Ioannis

Instructions for preparing your revised manuscript

1. When you are ready to submit the revision, please upload:

- A Word file of the manuscript text (including legends of main Figures, EV Figures and Tables). Please make sure that changes are highlighted (or "tracked") to be clearly visible.

- Individual production-quality figure files (one file per figure). When assembling your figures, please refer to our figure preparation guidelines in order to ensure proper formatting and readability in print as well as on screen:

If the data shown in a figure are obtained from n {less than or equal to} 2, please use scatter plots showing the individual data points.

i. the name of the statistical test used to generate error bars and P values

ii. the number (n) of independent experiments (please specify technical or biological replicates) underlying each data point

(discussion of statistical methodology can be reported in the Materials and Methods section, but figure legends should contain a

basic description of n, P, and the test applied)

iii. the nature of the bars and error bars (s.d., s.e.m.).

- A point-by-point response to the referees' comments, with a detailed description of the changes made (as a word file). All referees' concerns must be fully addressed and their suggestions taken on board. When preparing your letter of response to the referees' comments, please bear in mind that this will form part of the Review Process File and will therefore be available online to the community. Please note that you have the possibility to opt out of the transparent process at any stage prior to publication by letting the editorial office know (contact@embojournal.org); if you do opt out, the Review Process File link will point to the following statement: "No Peer Review File is available with this article, as the authors have chosen not to make the review process public in this case.". For more details on our Transparent Editorial Process, please visit our website:

<https://www.embopress.org/page/journal/14602075/authorguide#transparentprocess>

- Expanded View (EV) files (replacing Supplementary Information) that are collapsible/expandable online. A maximum of 5 EV Figures can be typeset. EV Figures should be cited as "Figure EV1, Figure EV2" etc. in the text, and their respective legends should be included in the manuscript file after the legends of regular figures. See detailed instructions regarding Expanded View files here:

- For the figures that you do NOT wish to display as Expanded View figures, they should be bundled together with their legends in a single PDF file called "Appendix", which should start with a short Table of Contents (including page numbers). Appendix figures should be referred to in the main text as: "Appendix Figure S1, Appendix Figure S2" etc. Please see detailed instructions here: <https://www.embopress.org/page/journal/14602075/authorguide#expandedview>

- A complete author checklist, which you can download from our author guidelines

(<https://www.embopress.org/page/journal/14602075/authorguide>). Please note that the checklist will also be part of the Review Process File.

2. Please note that no statistics should be calculated and shown in Figures if $n=2$. Please also note that each p value should be reported as an exact value.

3. Before submitting your revision, primary datasets (and computer code, where appropriate) produced in this study need to be deposited in appropriate public databases (see <https://www.embopress.org/page/journal/14602075/authorguide#dataavailability>). In particular, we kindly ask you to deposit the RNA sequencing data produced in your study. Their accession numbers, databases, and the specific URLs (links) should be listed in a formal "Data availability" section (placed after Methods).

*** The Data Availability Section is restricted to new primary data that are part of this study. In case you have no data that require deposition in a public database, please state so instead of referring to the database: "Our study includes no data deposited in public repositories." under the heading "Data availability". ***

*** All links should resolve to a page where the data can be accessed. ***

*** Please remember to provide in the Data availability section of your revised manuscript reviewer passwords if the datasets are not yet public. ***

*** Please use detailed data citations for already available datasets that were re-analyzed in your study - for more information on the format, see point #9 below. ***

4. Please check that the title and the abstract of the manuscript are brief, yet explicit, even to non-specialists. The length of the title should not exceed 100 characters, and the abstract should be a single paragraph not exceeding 175 words.

5. All materials and methods need to be described in the manuscript using our "Structured Methods" format, which is now required for all research articles. According to this format, the Methods section includes a single "Reagents and Tools Table" - listing key reagents, experimental models, software and relevant equipment including their sources and relevant identifiers- followed by a "Methods and Protocols" section describing the methods. Please download and fill our Reagents and Tools Table template (.docx), which you can find in our author guide:

<https://www.embopress.org/page/journal/14602075/authorguide#structuredmethods>. When submitting your revised manuscript, please do not include the Reagents and Tools Table in the Methods section of the manuscript but upload it as a separate file choosing the file type "Reagent Table".

6. Please also note our reference format: <https://www.embopress.org/page/journal/14602075/authorguide#referencesformat>.

7. At EMBO Press we ask authors to provide source data for the main manuscript figures. Our source data coordinator will contact you to discuss which figure panels we would need source data for and will also provide you with helpful tips on how to

upload and organize the files.

8. Please remember: digital image enhancement is acceptable practice, as long as it accurately represents the original data and conforms to community standards. If a figure has been subjected to significant electronic manipulation, this must be noted in the figure legend or in the "Materials and Methods" section. The editors reserve the right to request original versions of figures and the original images that were used to assemble the figure.

9. Our journal encourages inclusion of data citations in the reference list to directly cite datasets that were obtained from public databases. Data citations in the article text are distinct from normal bibliographical citations and should directly link to the database records from which the data can be accessed. In the main text, data citations are formatted as follows: "Data ref: Smith et al, 2001" or "Data ref: NCBI Sequence Read Archive PRJNA342805, 2017". In the Reference list, data citations must be labeled with "[DATASET]". A data reference must provide the database name, accession number/identifiers, and a resolvable link to the landing page from which the data can be accessed at the end of the reference. Further instructions are available at: <https://www.embopress.org/page/journal/14602075/authorguide#referencesformat>.

10. We request authors to consider both actual and perceived competing interests. Please review our policy (<https://www.embopress.org/page/journal/14602075/authorguide#conflictsofinterest>) and update your competing interests statement if necessary. Please name this section 'Disclosure and competing interests statement' and place it after the Acknowledgements section.

11. Please note that all corresponding authors are required to provide an ORCID ID upon submission of a revised manuscript (<https://orcid.org/>). Please find instructions on how to link your ORCID ID to your account in our manuscript tracking system in our Author guidelines (<https://www.embopress.org/page/journal/14602075/authorguide#authorshipguidelines>).

12. We use CRediT to specify the contributions of each author in the journal submission system. CRediT replaces the author contribution section, which should be removed from the manuscript. Please use the free text box to provide more detailed descriptions. See also guide to authors: <https://www.embopress.org/page/journal/14602075/authorguide#authorshipguidelines>.

14. We would also welcome the submission of cover suggestions or motifs to be used by our Graphics Illustrator in designing a cover.

15. Please use the link below to submit your revision:
<https://emboj.msubmit.net/cgi-bin/main.plex>

Arbitrator:

This is not a full review but rather my overall assessment based on the invitation to arbitrate:

Overall, the study by Sala et al is interesting and technically of high quality.

While the work presents valuable insights, I agree with the reviewers that the conceptual advancement remains modest. The suggested revisions would improve the manuscript, particularly those aimed at elucidating the mechanism by which IFN γ inhibits Tfh development and expanding the study to later timepoints will be useful. In addition, the authors should give stronger credit to the existing literature as pointed out by reviewer 1. Particularly, some of the arguments the authors raise in response to the comments of Reviewer 1 are not fully substantiated by data presented in their study (e.g. that a direct competition exists between Th1 and Tfh cells - this is not experimentally addressed, that the Tcf1+ population is distinct from the Th1 precursor - this would require transfers of sorted populations and follow-up analysis). If these points are fully addressed, the manuscript could be suitable for publication in EMBO Journal.

Dear Ioannis,

we thank you for considering our manuscript for publication in *The EMBO Journal* and we are happy to be able to submit a revised version addressing the questions and the concerns of the arbitrator and the reviewers. We have highlighted all the changes in yellow in the manuscript. All in all, we have added 7 new figures (1 Main and 6 Supplementary) and modified 12 figures (6 Main and 6 Supplementary).

Please find below a detailed response to the Arbitrator, and further a point-by-point rebuttal to the reviewers.

Arbitrator:

This is not a full review but rather my overall assessment based on the invitation to arbitrate: Overall, the study by Sala et al is interesting and technically of high quality.

We thank the arbitrator for the overall positive assessment of our work.

While the work presents valuable insights, I agree with the reviewers that the conceptual advancement remains modest. The suggested revisions would improve the manuscript,

We have addressed the reviewer's comments and concerns and are happy to provide a revised version of the manuscript, which in our view is greatly improved. Specifically:

- 1. We have investigated the mechanisms by which IFN- γ inhibits T_{FH} development and we have found that IFN- γ is produced by T cells (which is now also reflected in the title of the paper) but is not sensed by CD4⁺ T cell themselves; rather IFN- γ targets another cell type in the dLN microenvironment, likely DCs (Fig. 7). Indeed we are able to show that DC upregulate the IFN γ transcriptional signature 48 hrs after LCMV infection(Fig. S15A). In addition, leveraging previously published data, we show that IFN- γ sensing modifies DC transcriptional profile (Fig. S15B). Due to time constraints, we could not perform functional studies to formally validate our hypothesis, which would require the importing of new mouse models and would take almost another year of work. While these results remain correlative, we believe they provide a strong foundation for future mechanistic studies.*
- 2. We have extended the study at later time-points as requested (Fig. 5 and 8), and improved characterization of the T_{FH} phenotype acquired by Tcf-1⁺ cells (Fig. S6C).*
- 3. We have added a new infection model, aerosol SARS-CoV-2 infection, where we confirm the role of IFN- γ in T_{FH} suppression.*
- 4. We have addressed reviewers' major and minor comments by adding the requested gating strategies and absolute numbers (see specific responses)*
- 5. We have cited and discussed all the references suggested by the reviewers.*
- 6. We have substantially modified abstract, introduction and discussion sessions to better reflect our findings*

particularly those aimed at elucidating the mechanism by which IFN γ inhibits Tfh development and expanding the study to later timepoints will be useful.

As suggested by both the arbitrator and the reviewers we have performed experiments aiming at elucidating the mechanism by which IFN- γ inhibits T_{FH} development. Notably, by using genetic models and Crispr/Cas9 technology, we found that the IFN- γ responsible for T_{FH} suppression is T cell-derived but does not act on CD4⁺ T cells themselves. Instead, IFN- γ is sensed by DCs, which might then acquire a phenotype that interferes with T_{FH} differentiation. This is quite novel, since previous studies reporting a role for IFN- γ in T_H1 maintenance have suggested an autocrine mechanism (Bradley et al., 1996; Lighvani et al., 2001; Whitmire et al., 2005). The new data are reported in New Figures 7 and S16.

With regard to the second request “expanding the study to later timepoints will be useful” we have added new data on Smarta CD4⁺ T cell polarization 14 days after infection in Figure 5, panels I-K. As discussed in the text, “The frequency of Smarta cells at this time-point was highly variable and significantly lower than day five, due to the contraction phase all T cells undergo after viral clearance (Fig. 5J). Nonetheless, IFN- γ blockade still resulted in increased T_{FH} frequencies (Fig. 5K) indicating that a single-dose of blocking antibody at the beginning of infection was sufficient to affect T cell polarization for weeks.” Unfortunately, a substantial effect at later time-points was not observed in GC B cells, probably due to the fact that LCMV adopts many different mechanisms to interfere with GC responses (Kuka and Iannacone, 2018). In another setting, two weeks after Sars-CoV-2 infection, we observed a trend in the increase of IgG1 and IgG3 antibodies, which have been reported to be the only TFH-dependent isotypes (Chen et al., 2022) (Fig. 8G).

In addition, the authors should give stronger credit to the existing literature as pointed out by reviewer 1.

As discussed in the below point-by-point rebuttal we have cited and discussed all papers suggested by the reviewers.

Particularly, some of the arguments the authors raise in response to the comments of Reviewer 1 are not fully substantiated by data presented in their study (e.g. that a direct competition exists between Th1 and Tfh cells - this is not experimentally addressed,

Previous work has shown some degree of overlap or competition between T_{FH} and T_{H1} subsets (Nakayamada et al., 2011; Lönnberg et al., 2017; Weinstein et al., 2018). For example it was reported that T_{FH} and T_{H1} share a transitional phase expressing both T-bet and Bcl-6. While the cells progress into reinforcing T_{H1} phenotype, T-bet suppresses further T_{FH} differentiation by competing with Bcl-6. This competition seems to be cell-intrinsic since CD4⁺ T cells lacking T-bet differentiate into T_{FH} (Nakayamada et al., 2011; Lönnberg et al., 2017). We propose that T_{H1} antagonize T_{FH} not only through competition of cell-intrinsic transcription factors as proposed by others {Nakayamada et al., 2011 #64361}, but also through a cell-extrinsic effect on the surrounding microenvironment.

that the Tcf1+ population is distinct from the Th1 precursor - this would require transfers of sorted populations and follow-up analysis).

The proposed experiment would require that we sort effector Tcf-1 cells at day 5 upon infection and transfer them into recipients that will be reinfected. Unfortunately these effector cells cannot enter the LNs (at least not in sufficient amounts) because they lack CD62L, which is instead expressed by naive T cells. Due to this caveat, we have always been resistant in doing this kind of experiment in our setting. However, upon the arbitrator’s request, we tried to do so. We sorted Slamf6⁺ cells (proxy for Tcf-1) and CXCR6⁺ cells (proxy for GzmB⁺ cells) from LNs pooled from 20 LCMV-infected mice. The purity of the cells after sorting was very high, and we were able to transfer 250000 cells for each recipient mice, which was then infected with LCMV and analyzed 3 days after. As expected, we found very few Smarta cells in the LNs. Smarta of mice that received Slamf6⁺Tcf-1⁺ cells were characterized by a predominant Tcf-1⁺ phenotype, suggesting that they maintain their phenotype and do not give rise to similar proportions of GzmB and Tcf-1 cells as naive T cells do.

However, the number of Smarta is so low that we don’t feel confident in publishing these data.

Dear Mirela, dear Matteo,

Thank you again for submitting your substantially revised manuscript (EMBOJ-2024-119046R) to The EMBO Journal, and for your patience during re-review. Your manuscript has now been seen by the arbitrator who had also assessed the previous version, and we have received their comments, which you can find below.

I am very pleased to say that the arbitrator finds the manuscript thoroughly revised and substantially improved, now including further data on the mechanism by which IFN-gamma inhibits T follicular helper cell development. While he/she recognizes that the majority of the previously raised concerns have been successfully addressed, two limitations regarding the new data/conclusions are pointed out, namely a possible discrepancy in the data supporting the conclusion that IFN-gamma does not act on CD4+ T cells, and the lack of conclusive evidence proving that dendritic cells are indeed the cell population that senses IFN-gamma.

In light of this input, I would like to invite you to address these two remaining points in a final version of your manuscript. Regarding the first point, the statistical significance of the difference must be provided, and the apparent discrepancy must be clarified and discussed. Regarding the second point, although we agree with the arbitrator that the work and the manuscript would benefit from the addition of experimental data proving that dendritic cells sense IFN-gamma, such data will not be required for the publication of the manuscript in The EMBO Journal. The limitation must, however, be discussed and the conclusions appropriately toned down in the absence of conclusive experimental data. Please include in your resubmission a detailed point-by-point response letter fully addressing both points.

There are also a few editorial requests/formatting changes that we need from you to address in the final version of your manuscript, before we can proceed with formal acceptance of the manuscript for publication in The EMBO Journal. Please briefly describe how they are addressed in a brief cover letter:

- We noticed a discrepancy in the name of one of your co-authors that should be resolved before publication of the manuscript: Jia Nie in the manuscript vs. Nie Jia in the author's profile in our manuscript tracking system. Please make sure that the name is provided correctly both in the authors' list in the manuscript and in the author's profile.
- Please change the heading "Data and materials availability" to "Data availability". The reviewer tokens can now be removed from this section, while you are kindly requested to make sure that all deposited datasets will be publicly available at the time of publication. Please include in this statement the permanent URLs (links) to all deposited datasets. I would like to mention that this section should contain information only about the new datasets that were produced during the study. Previously generated datasets that were re-analyzed in the study should not be included in the Data availability statement; instead, our journal encourages inclusion of data citations in the reference list to directly cite datasets that were obtained from public databases. Data citations in the article text are distinct from normal bibliographical citations and should directly link to the database records from which the data can be accessed. In the main text, data citations are formatted as follows: "Data ref: Smith et al, 2001" or "Data ref: NCBI Sequence Read Archive PRJNA342805, 2017". In the Reference list, data citations must be labeled with "[DATASET]". A data reference must provide the database name, accession number/identifiers, and a resolvable link to the landing page from which the data can be accessed at the end of the reference. Further instructions are available at: <https://www.embopress.org/page/journal/14602075/authorguide#referencesformat>.
- Please change the heading "Conflicts of interests disclosure" to "Disclosure and competing interests statement".
- The author contributions statement should be removed from the manuscript file. Instead, we use CRediT to specify the contributions of each author in the journal submission system. Please feel free to use the free text box to provide more detailed descriptions during submission. See also our guide to authors for more information: <https://www.embopress.org/page/journal/14602075/authorguide#authorshipguidelines>.
- Please note that "et al." should be used after the names of the first 10 co-authors for publications with more than 10 co-authors in the References list. You can find more information about our citation format in our guide to authors: <https://www.embopress.org/page/journal/14602075/authorguide#referencesformat>.
- Please rename "Materials and Methods" to "Methods".
- Please move the information regarding experiments involving animals from your Author Checklist to the Methods section of your manuscript. In the last column of your Author Checklist, please only indicate in which section of the manuscript the information can be found (in "Methods", in this case). Please make sure to provide the details of the authority granting ethics approval (including the reference number) of these experiments, and also include a statement of compliance with ethical regulations.
- Please note that the full funding information must be provided both in the Acknowledgements section of the manuscript and in

our online manuscript tracking system (eJP) during resubmission. Please resolve the following inconsistencies: 2017ZXT5WR is provided in the manuscript vs. 2017ZXT5W in eJP; missing in eJP: 22737, 12365801, Gilead Sciences, Asher Biotherapeutics and VIR Biotechnology, Fondazione Prossimo Mio, FONDAZIONE SAME.

- We noticed that callouts are missing for Fig. 2J, Fig. 8H. Please make sure that all Figure panels are called out in your revised manuscript.

- Please rename your Tables 1-3 to "Dataset EV1-EV3". Their legends must be removed from the main manuscript file and instead be provided in a separate tab of each corresponding Excel file.

- Please add the heading "Appendix for" followed by the manuscript title on the first page of your Appendix PDF file. There is no need for the authors' list and affiliations to be included in this file. Instead, please include a brief Table of Contents (including page numbers) on the first page(s) of your Appendix file. The nomenclature of the Appendix figures should be corrected to "Appendix Figure S1-S17" throughout the Appendix file and the manuscript (please update all callouts accordingly). Each figure legend in the Appendix file should be provided right after the corresponding Figure. Please do not use "Supplementary figure legends".

- The materials and methods need to be described in the manuscript using our structured methods format, which is now required for all research articles. According to this format, the Methods section includes a single "Reagents and Tools Table" -listing key reagents, experimental models, software and relevant equipment including their sources and relevant identifiers- followed by a "Methods and Protocols" section describing the methods. Please download and fill our Reagents and Tools Table template (.docx), which you can find in our author guide:

<https://www.embojournal.org/page/journal/14602075/authorguide#structuredmethods>. When submitting your revised manuscript, please do not include the Reagents and Tools Table in the Methods section of the manuscript but instead upload it as a separate file choosing the file type "Reagent Table".

- Thank you for depositing the source data for the original Figure panels. While checking them, we noticed that no source data have been deposited for the new Figure panels that were added during revision of the manuscript. Could you please add the new source data for Figure panels 4C, 4F and 4G; 5J and 5K; 6C, 6D and 6E; 7B-J; 8F and 8G?

- During our routine pre-acceptance checks, our data editors have raised the following queries:

"Please provide the exact p values in the legends of Figures 1G, I, J, K; 2B, D, F; 3B, C; 4B, D, F, G; 5B, C, D, F, G, H, K; 6C, D, E; 7D, G, H, J; 8B, C, D, F, I, J."

- The manuscript section order should be corrected as follows: Title page - Abstract & Keywords - Introduction - Results - Discussion - Methods - Data Availability - Acknowledgements - Disclosure and Competing Interests Statement - References - Figure Legends - main Table(s) with legends (if there are any) - Expanded View Figure Legends.

- Please note that EMBO press papers are accompanied online by:

A) a short (2 sentences) summary of the findings and their significance,

B) 2-5 short bullet points highlighting the key results, and

C) a synopsis image in .jpg or .png format that is exactly 550 pixels wide and 300-600 pixels high (the height is variable). Please note that the text needs to be legible at the final size.

Please upload this information along with your revised manuscript (the text for A and B should be provided in a separate Word file).

Please also note that as part of the EMBO publications' Transparent Editorial Process, The EMBO Journal publishes online a Peer Review File along with each accepted manuscript. This File will be published in conjunction with your paper and will include the referee reports, your point-by-point response and all pertinent correspondence relating to the manuscript. You can opt out of this by letting the editorial office know (contact@embojournal.org). If you do opt out, the Peer Review File link will point to the following statement: "No Peer Review File is available with this article, as the authors have chosen not to make the review process public in this case."

We look forward to seeing a final version of your manuscript as soon as possible. Please let us know if you have any questions and use this link to submit your revision: <https://emboj.msubmit.net/cgi-bin/main.plex>.

Best wishes,

Ioannis

Ioannis Papaioannou, PhD

Arbitrator:

The authors present a substantially revised and clearly improved manuscript, which appropriately credits previous findings. They now include data from later timepoints and have further investigated the underlying mechanism by which IFN γ inhibits Tfh development. Their additional data show that IFN γ doesn't act directly on CD4 $^+$ T cells but likely acts on DCs.

While most points have been addressed, some issues remain regarding the experiments aimed at elucidating the mechanism.

1. Some questions remain regarding the conclusion that IFN γ doesn't act on CD4 $^+$ T cells. In Fig 7E no difference in Tfh vs Th1 proportions is observed (which supports the conclusion) but in Fig 7H it looks like there is a 2-fold difference in the proportion of Tfh (comparing the control groups=not treated with the Ab), albeit no decrease in Th1 proportion. Is this difference statistically significant and how do the authors explain this discrepancy?

2. While the authors conclude that DCs are likely the cell population that senses IFN γ , they do not test this experimentally. This should be done to conclude the mechanistic analysis of this process (eg with mixed bm chimeras IFN γ KO : CD11c-DTR).

All other points have been addressed.

February 21, 2025

Point-by-point rebuttal

Dear Mirela, dear Matteo,

Thank you again for submitting your substantially revised manuscript (EMBOJ-2024-119046R) to The EMBO Journal, and for your patience during re-review. Your manuscript has now been seen by the arbitrator who had also assessed the previous version, and we have received their comments, which you can find below.

I am very pleased to say that the arbitrator finds the manuscript thoroughly revised and substantially improved, now including further data on the mechanism by which IFN-gamma inhibits T follicular helper cell development. While he/she recognizes that the majority of the previously raised concerns have been successfully addressed, two limitations regarding the new data/conclusions are pointed out, namely a possible discrepancy in the data supporting the conclusion that IFN-gamma does not act on CD4+ T cells, and the lack of conclusive evidence proving that dendritic cells are indeed the cell population that senses IFN-gamma.

In light of this input, I would like to invite you to address these two remaining points in a final version of your manuscript. Regarding the first point, the statistical significance of the difference must be provided, and the apparent discrepancy must be clarified and discussed. Regarding the second point, although we agree with the arbitrator that the work and the manuscript would benefit from the addition of experimental data proving that dendritic cells sense IFN-gamma, such data will not be required for the publication of the manuscript in The EMBO Journal. The limitation must, however, be discussed and the conclusions appropriately toned down in the absence of conclusive experimental data. Please include in your resubmission a detailed point-by-point response letter fully addressing both points.

Dear Ioannis,

We are happy that the thorough revision of our manuscript was appreciated by both the editorial board and the arbitrator. Please find below our response to the last two points raised by the arbitrator.

Arbitrator:

The authors present a substantially revised and clearly improved manuscript, which appropriately credits previous findings. They now include data from later timepoints and have further investigated the underlying mechanism by which IFN γ inhibits Tfh development. Their additional data show that IFN γ doesn't act directly on CD4+ T cells but likely acts on DCs.

We thank the arbitrator for the positive assessment of our revised manuscript and we agree that the manuscript is now highly improved.

While most points have been addressed, some issues remain regarding the experiments aimed at elucidating the mechanism.

1. Some questions remain regarding the conclusion that IFN γ doesn't act on CD4 $^+$ T cells. In Fig 7E no difference in T_{fh} vs T_{H1} proportions is observed (which supports the conclusion) but in Fig 7H it looks like there is a 2-fold difference in the proportion of T_{fh} (comparing the control groups = not treated with the Ab), albeit no decrease in T_{H1} proportion. Is this difference statistically significant and how do the authors explain this discrepancy?

We had already noticed these slight differences between the groups mentioned by the arbitrator. Although we don't have a formal explanation, we think these little variations might be indirectly linked to the fact that CD4 T cells unable to sense IFN- γ might be able to produce less of it (due to a feedback mechanism). However these slight differences are not statistically significant, as shown in the graph below. This is the reason why we had already stated in the text "*However, no significant differences were observed in the polarization between WT and IFNGR1 $^{-/-}$ cells*" in lines 311-312. We have now updated these figures showing all statistically significant differences and mentioned in the methods section that **comparisons are not statistically significant unless indicated**. We hope these actions will help clarify the issue.

2. While the authors conclude that DCs are likely the cell population that senses IFN γ , they do

no not test this experimentally. This should be done to conclude the mechanistic analysis of this process (eg with mixed bm chimeras IFNGR KO : CD11c-DTR).

This is a fair point and, as suggested by the editor, we have toned down this conclusion in several points of the manuscript (all changes are highlighted in yellow):

- In the Abstract, line 40, we have substituted “likely” with “possibly” (we could not due more due to space constraints)
- In the Results, lines 330-331, we have again substituted “likely” with “possibly”, and we have added the sentence “*However, further functional studies are required to corroborate this hypothesis*”
- In the Discussion, lines 386-387, we have added “*Although the exact cell type targeted by IFN- γ should be further corroborated by functional studies, ...*”
- In the synopsis figure, we have added other cell types in addition to DC

We hope that these changes will clarify that the hypothesis that IFN- γ might be sensed by DC to suppress T_{FH} differentiation has still to be functionally corroborated, although it is based on solid preliminary data.

All other points have been addressed.

Dear Mirela, dear Matteo,

Congratulations on an excellent work! I am very pleased to inform you that your manuscript has now been accepted for publication in The EMBO Journal. Thank you very much for your thorough responses to the referees' and arbitrator's concerns, and for addressing our editorial and formatting requests.

If you have any questions, please do not hesitate to contact the Editorial Office. Thank you for your contribution to The EMBO Journal. Working with you has been a pleasure!

Best regards,

Ioannis
